# Unique universal scaling in nanoindentation pop-ins

Yuji Sato [1], Shuhei Shinzato[1], Takahito Ohmura[2,3,4 ✉], Takahiro Hatano[5 ✉] & Shigenobu Ogata [1,3 ✉]

Power laws are omnipresent and actively studied in many scientific fields, including plasticity of materials. Here, we report the power-law statistics in the second and subsequent pop-in magnitudes during load-controlled nanoindentation testing, whereas the first pop-in is characterized by Gaussian-like statistics with a well-defined average value. The transition from Gaussian-like to power-law is due to the change in the deformation mechanism from dislocation nucleation to dislocation network evolution in the sharp-indenter induced abruptly decaying stress and dislocation density fields. Based on nanoindentation testing on the (100) and (111) surfaces of body-centered cubic (BCC) iron and the (100) surface of face-centered cubic (FCC) copper, the scaling exponents of the power laws were determined to be 5.6, 3.9, and 6.4, respectively. These power-law exponents are much higher than those typically observed in micro-pillar plasticity (1.0–1.8), suggesting that the nanoindentation plasticity belongs to a different universality class than the micro-pillar plasticity.

[1] Department of Mechanical Science and Bioengineering, Graduate School of Engineering Science, Osaka University, 1-3 Machikaneyama, Toyonaka, Osaka 560-8531, Japan. [2] Research Center for Structural Materials, National Institute for Materials Science (NIMS), 1-2-1 Sengen, Tsukuba, Ibaraki 305-0047, Japan. [3] Center for Elements Strategy Initiative for Structural Materials (ESISM), Kyoto University, Yoshida Honmachi, Sakyo, Kyoto 606-8501, Japan. [4] Graduate School of Engineering, Kyushu University, 744 Motooka, Nishi-ku, Fukuoka 819-0395, Japan. [5] Department of Earth and Space Science, Graduate School of Science, Osaka University, 1-1 Machikaneyama, Toyonaka, Osaka 560-0043, Japan. ✉email: OHMURA.Takahito@nims.go.jp; hatano@ess.sci.osaka-u.ac.jp; ogata@me.es.osaka-u.ac.jp

**P**ower laws are ubiquitous and actively studied in many fields of science, especially in statistical studies of the magnitudes of natural phenomena such as earthquakes[1]. They are also observed in the plasticity of micro- and nanoscale materials in mechanical testing[2–10]. For many years, nanoindentation has been widely used in fundamental studies of the local strength and plasticity of materials[11]. It is well-known that a catastrophic event, called the displacement burst or "pop-in," is observed during load-controlled nanoindentation[12]. During the first pop-in, a high-density dislocation network is formed right beneath the indenter[13], because a large amount of elastic energy can be stored before the first pop-in due to the lack of mobile defects that can release the stored elastic energy by generating plastic strain. Thus, upon a large-scale catastrophic event of plasticity, i.e., the first pop-in, homogeneous[14,15] or heterogeneous dislocation nucleation from the immobile defects[16,17] triggers the release of the stored elastic energy. In general, after the first pop-in, smaller pop-ins (second and subsequent pop-ins) occur intermittently with further evolution of the dislocation network via dislocation avalanches. Such intermittent events due to dislocation avalanches have also been observed as serrated stress–strain curves in displacement-controlled uniaxial-loading pillar-compression testing at micro- and submicron scales. The magnitude of stress drops in these experiments exhibits a power law within a certain range[6,18]. Although nanoindentation- and pillar-compression testing have different boundary conditions and stress distributions in the target material, the intermittent pop-ins may obey power-law statistics because both of them are driven by dislocation avalanches. Recently, it was suggested that the pop-in magnitude also obeys a power law for nanoindentation testing by simulation[19] and experiment[20]. The power-law exponent was estimated to be around 1.5–1.6 for face-centered cubic (FCC) metals. However, to the best of our knowledge, the background of the power-law exponent in nanoindentation testing is still unclear.

In this study, nanoindentation experiments are conducted on the ⟨100⟩- and ⟨111⟩-oriented surfaces ((100) and (111) surfaces) of body-centered cubic (BCC) iron (Fe) and the (100) surface of FCC copper (Cu) at room temperature (300 K). Then, stochastic analyses of the pop-in magnitude are performed by defining the pop-in magnitude as the indenter displacement burst and as the drop of the contact stress between the indenter and target materials.

## Results

**Indentation load-displacement curve.** Figure 1 shows a typical indentation load ($P$)-displacement ($h$) curve for a Fe single crystal for indentations made normal to the sample surface (100). The testing temperature was 300 K, and the loading rate was 50 μNs$^{-1}$. The curve shows that two types of pop-in events occurred in the loading sequence, such as the first pop-in and the second and subsequent pop-ins. The first pop-in, which is indicated by a dark-blue solid arrow, was unique: it had the largest magnitude in terms of the displacement burst, $\Delta h$, of the pop-ins observed in each nanoindentation testing. The detected second and subsequent pop-ins with a criterion we state later, which are indicated by red solid arrows, were of a different type compared with the first pop-in: they were much smaller in $\Delta h$ compared with the first pop-in, as shown by the magnified plot inset in Fig. 1. A set of measured load ($P$) and displacement ($h$) data was recorded at a rate of 200 measurements per second (i.e., with an interval of 5 ms).

It is difficult to discriminate between a real pop-in signal and electrical and mechanical noise. Therefore, a threshold value, $\Delta h^c$, above which the displacement burst is considered to be a pop-in, was set such that only the $\Delta h$s satisfying $\Delta h = h(t + n\Delta t) - h(t) > \Delta h^c$ were used for the subsequent stochastic analyses, where $h(t)$ is the displacement at time $t$, $n$ determines the limit of an acceptable load change $\Delta P^c$ during a single displacement burst (ideally zero), and $\Delta t$ is an interval time of measurements ($\Delta t = 5$ ms) in our indentation experiments. To check the threshold dependency, three thresholds of $\Delta h^c = 0.5, 0.75,$ and $1.0$ nm were tested. Because it was found that the choice of thresholds did not change the trend in the results of the following stochastic analysis, the minimum value of 0.5 nm was simply used. Further verification of the threshold is discussed in Supplementary Note 1. We set $n = 2$, thus $\Delta P^c = P(t + 2\Delta t) - P(t) \approx 2\dot{P}\Delta t = 0.5$ μN, which is just above the force measurement limit ~0.3 μN.

**Probability distributions of pop-in magnitudes.** Using the pop-in data set, $\{P_{ij}^{\text{pop-in}}, h_{ij}, \Delta h_{ij}\}(i = 1, ..., N, \ j = 1, ..., n_i)$, obtained from $N = 1000$ indentations where $i$th indentation has $n_i$ (normally less than 10) pop-ins; the distributions of the event occurrence probability $p$ were plotted for the (100) and (111) surfaces in BCC Fe, and the (100) surface in FCC Cu, as shown in Fig. 2. Again, the testing temperature was 300 K, and the loading rate was 50 μNs$^{-1}$. The distributions were plotted with respect to the contact stress drop $\Delta\sigma$, which is defined as follows:

$$\Delta\sigma = \frac{P^{\text{pop-in}}}{A(h)} - \frac{P^{\text{pop-in}}}{A(h + \Delta h)}, \tag{1}$$

where $P^{\text{pop-in}}$ is the indentation load when a pop-in occurs, and $A(h)$ is the cross-sectional contact area of the indenter at an indentation depth of $h$. Thus, $P^{\text{pop-in}}/A(h)$ represents the average contact stress (pressure) between the indenter and the target material at an indentation depth of $h$. For the first pop-in, Heltz's contact theory[21] was assumed to describe the cross-sectional contact area between the spherical indenter tip and a flat material surface: $A(h) = \pi a^2 = \pi R h$, where $a = \sqrt{Rh}$ is the contact radius and $R$ is the radius of the indenter tip. This assumption is based on the idea that, before the first pop-in, only the indenter tip, which can be assumed to have a spherical shape, touches the material surface. A molecular dynamics (MD) nanoindentation simulation actually demonstrates the validity of the assumptions in the contact area $A(h) = \pi a^2 = \pi R h$ (see "Methods" and Supplementary Note 2). The contact area at the end of the first pop-in, such as $A(h + \Delta h)$, and at the beginning and the end of the second and subsequent pop-ins, such as $A(h)$ and $A(h + \Delta h)$

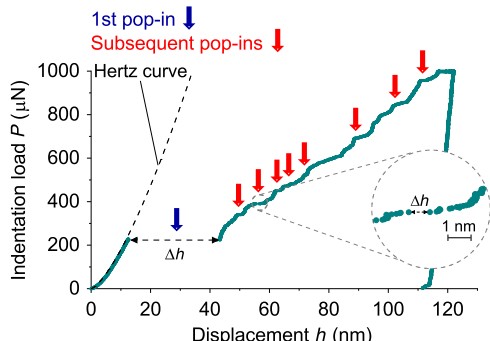

**Fig. 1 Typical indentation load ($P$) vs. displacement ($h$) curve for the (100) surface of BCC Fe single crystal.** The green line indicates the indentation load-displacement curve. The blue arrow and the red arrows indicate the first pop-in and subsequent pop-in events, respectively, in the indentation. The curved broken line is drawn by Hertz's contact theory fitted with the indentation load-displacement data before the first pop-in.

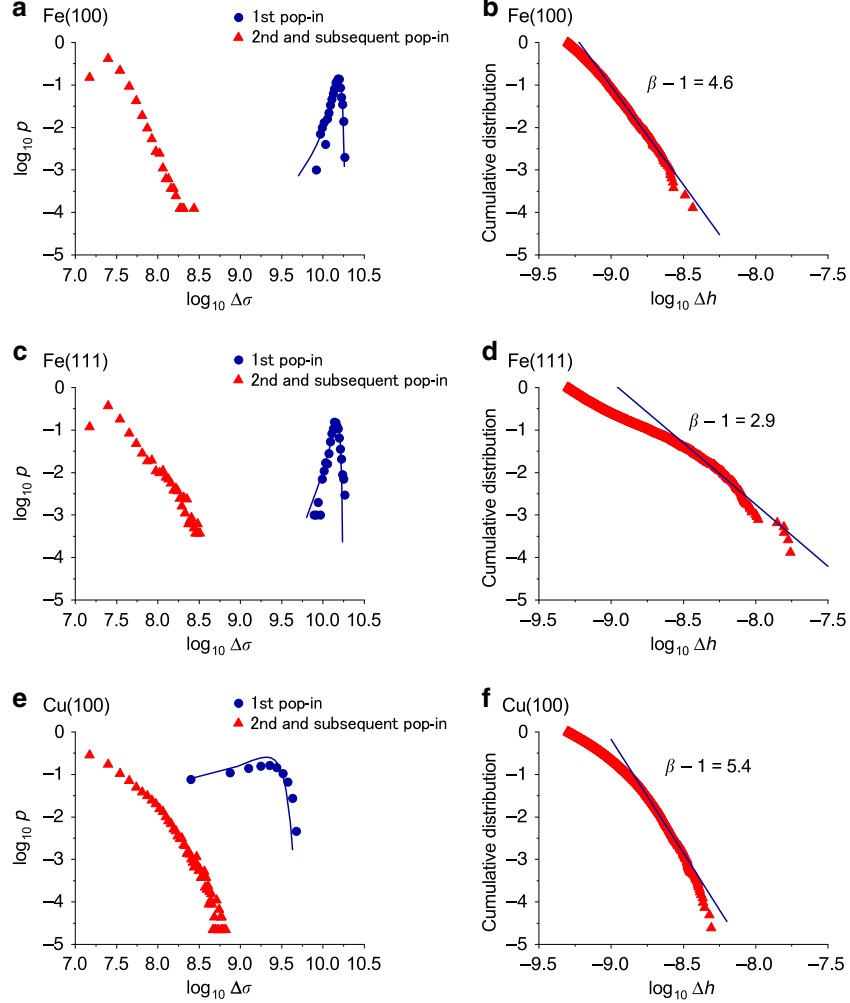

**Fig. 2 Probability distributions of pop-in magnitudes. a, c, e** Probability distributions of the first pop-in and second and subsequent pop-in magnitudes as a function of the stress drop $\Delta\sigma$ (Pa) for the (**a**) (100) and (**c**) (111) surfaces of BCC Fe, and (**e**) the (100) surface of FCC Cu, obtained by equal-width binning ($5.0 \times 10^8$ Pa for the first pop-in and $1.0 \times 10^7$ Pa for the second and subsequent pop-ins). The solid line is drawn by using the theory, Eq. (2) (see text). **b, d, f** Probability distributions of subsequent pop-in magnitudes as a function of the displacement burst $\Delta h$ (m) for the (**b**) (100) and (**d**) (111) surfaces of BCC Fe, and (**f**) the (100) surface of FCC Cu, obtained by bin-free cumulative distribution. The testing temperature is 300 K and the loading rate is 50 μNs$^{-1}$. The power-law exponent is estimated by least-square fitting using the data within $-9.0 \leq \log_{10}\Delta h$ for BCC Fe (100), $-8.5 \leq \log_{10}\Delta h$ for BCC Fe (111), and $-8.8 \leq \log_{10}\Delta h$ for FCC Cu (100). See also Supplementary Note 5 for equal-width binning and logarithmic binning plots.

respectively, is estimated by a quadratic area function: $A(h) = A(h_{con}(h)) = c_2 (h_{con}(h))^2 + c_1 h_{con}(h) + c_{\frac{1}{2}}(h_{con}(h))^{\frac{1}{2}}$ ($c_2 = 24.5$, $c_1 = 2.61 \times 10^3$ nm, and $c_{\frac{1}{2}} = 1.57 \times 10^{-7}$ nm$^{\frac{3}{2}}$) considering the effect of the rounded indenter tip shape through the Oliver–Pharr (OP) method[22] (for details, see "Methods" and Supplementary Note 3). This is based on the idea that the material surface has a trigonal pyramid-like shape, reflecting the shape of the indenter body because the indenter tip had already made a deep indent to the material.

For the second and subsequent pop-ins, cumulative distributions were also plotted with respect to the displacement burst $\Delta h$, as shown in Fig. 2, and with respect to the stress drop $\Delta\sigma$, as shown in Supplementary Fig. 10 in Supplementary Note 4. For reference, the distributions of the second and subsequent pop-ins are also plotted using equal-width and logarithmic binning as shown in Supplementary Fig. 11 in Supplementary Note 5. The data show that the stress drop magnitudes for the first pop-in exhibit a Gaussian-like distribution with a well-defined average;

on the other hand, the displacement burst and stress drop magnitudes of the second and subsequent pop-ins follow a power-law distribution.

## Discussion

The results are consistent with the fact that the first pop-in is triggered by dislocation nucleation, which is driven by thermal activation[16,23]. In contrast, the second and subsequent pop-ins are dominated by the long- (indirect) and short-range (direct) dislocation interactions[13,24,25] in the dislocation network that is formed during the previous and even during the current pop-in. They are more driven by mechanical forces and thus less driven by thermal activation than the first pop-in.

The distribution of the first pop-in shown in Fig. 2 appears to be a Gaussian. However, it should not be purely Gaussian because the occurrence probability of a thermally activated process is described by an Arrhenius equation based on the transition-state theory[26]. In our recent study, the occurrence probability of the first pop-in event was formulated as a function of $P^{pop-in}$[23] (see

Supplementary Note 6 for details)

$$p(P^{\text{pop-in}}) = \frac{k(P^{\text{pop-in}}) \exp\left[-\dot{P}^{-1} \int_0^{P^{\text{pop-in}}} k(P) dP\right]}{\int_0^{P_c^{\text{pop-in}}} k(P^{\text{pop-in}}) \exp\left[-\dot{P}^{-1} \int_0^{P^{\text{pop-in}}} k(P) dP\right] dP^{\text{pop-in}}}, \quad (2)$$

where $k(P^{\text{pop-in}})$ is the dislocation nucleation rate, and $P_c^{\text{pop-in}}$ is the maximum pop-in load at which the cumulative pop-in event probability approaches 1, and $\dot{P}$ is the loading rate. The concept of the model is basically the same as that of the model of Schuh and Lund[27], while here the probability is more directly represented as the function of $P^{\text{pop-in}}$ instead of the local shear stress at the dislocation nucleation point. Equation (2) can be rewritten as a function of the contact stress drop $\Delta\sigma$ with the following assumption:

$$\Delta\sigma = \sigma_e^{\text{contact}} - \sigma_c^{\text{contact}} \approx CP^{\text{pop-in}\frac{1}{3}} - \sigma_c^{\text{contact}}, \quad (3)$$

where $\sigma_c^{\text{contact}}$ is the average contact stress at the end of the first pop-in; this assumption implies that the dislocation activity ceases under the contact stress, which is assumed to be constant here for a specific target material and surface orientation. Based on Hertz's contact theory, the average contact stress $\sigma_e^{\text{contact}}$ immediately before the first pop-in under the load of $P^{\text{pop-in}}$ is proportional to $P^{\text{pop-in}\frac{1}{3}}$: $\sigma_e^{\text{contact}} = CP^{\text{pop-in}\frac{1}{3}}$, where $C$ is a constant[21]. Equations (2) and (3) can be combined to formulate the occurrence probability of the first pop-in event as a function of the contact stress drop $p(\Delta\sigma)$. Then, the unknown parameters of Eqs. (2) and (3) were determined by fitting the equations to the experimental data for the first pop-in (see also Supplementary Note 6) as shown in Fig. 2a, c, e (solid line). The theory describes the experimental results very well. The first pop-in distribution was also plotted on a linear scale as shown in Supplementary Fig. 12 in Supplementary Note 7.

To demonstrate the thermal activation nature of the first pop-in and the validity of Eq. (2) in terms of temperature and loading-rate dependencies, we performed additional nanoindentation tests at higher temperatures (373 and 473 K) at the same loading rate of $50 \,\mu\text{Ns}^{-1}$ in addition to the original 300K tests, and at a lower loading rate of $5 \,\mu\text{Ns}^{-1}$ at the same temperature of 300 K in addition to the original $50 \,\mu\text{Ns}^{-1}$ tests on the (100) surface of BCC Fe. The details of the experimental setup of the tests performed at different temperatures are described in Supplementary Note 8. The temperature and loading-rate dependencies of the first pop-in stress drop distribution and fitted theoretical curves based on Eq. (2) are shown in Supplementary Figs. 13 and 14 in Supplementary Note 8, respectively, which clearly reveal that Eq. (2) predicts well the Gaussian-like distributions at different temperatures and loading rates. In addition, we performed displacement-controlled MD nanoindentation simulations at 5 and 500 K (60 simulations for each temperature) on the (100) surface of BCC Fe; the force drop and pop-in load distributions, fitted theoretical curves for the pop-in load distribution, and the load-displacement curves are shown in Supplementary Note 9, which also strongly supports the temperature dependencies and the theory.

The power-law distribution of the second and subsequent pop-ins is evidence of the catastrophic nature of these events, such as dislocation avalanches with a universal scaling nature. Usually many second and subsequent pop-ins at different indentation depths could be detected even in one load-displacement curve, such as $1 \le n_i < 10$, as shown in Fig. 1. Since we would use as much of them as possible for accelerating the second and subsequent pop-in sampling, we need a guarantee of the statistical independency of the pop-ins with respect to $h$. To confirm the statistical independency, we plotted $\langle\Delta h|h\rangle$ vs. $h$ and $\langle\Delta\sigma|h\rangle$ vs. $h$ using all of the pop-in data ($N \times n_i$ data), where $\langle\Delta h|h\rangle$ and $\langle\Delta\sigma|h\rangle$ are the averages of the displacement burst $\Delta h$ and the

stress drop $\Delta\sigma$ at a given indentation depth $h$, respectively, as shown in Supplementary Figs. 22 and 23 in Supplementary Note 10. It is clearly seen that $\Delta h$ shows a nice statistical independency within a wide range of indentation depth; (20.0–40.0) nm $\le h \le$ (90.0–145.0) nm, while $\Delta\sigma$ shows a weak but some indentation depth dependency as seen in Supplementary Fig. 23. Because of this reason, just in this case, we have decided to use $\Delta h$ for the subsequent pop-in power-law analyses that were observed within the statistically independent displacement range (40.0 nm $\le h \le$ 120.0 nm (Fe(100)), 40.0 nm $\le h \le$ 95.0 nm (Fe(111)), and 20.0 nm $\le h \le$ 145.0 nm (Cu(111))), even though $\Delta\sigma$ can acceptably demonstrate the power law, as seen in Supplementary Fig. 10. We fitted the cumulative distribution of $\Delta h$ with a power-law function: $p(x) = \alpha x^{-\beta}$ as shown in Fig. 2, where $\alpha$ and $\beta$ are constants. The scaling exponents (power-law exponents) were estimated to be $\beta \sim 5.6$ for Fe(100), $\sim 3.9$ for Fe(111), and $\sim 6.4$ for Cu(100). All the obtained $\beta$s values are much larger than those that are typical of the pillar-compression testing, regardless of BCC, FCC, or metallic glass and loading direction (crystal orientation) $\beta = 1.0$–1.8[4–10]. Moreover, these are much higher than even those estimated in nanoindentation experiments and simulation for FCC Al and FCC Cu, $\beta \sim 1.6$[19,20].

The fundamental question, that is, why is there a difference in the power-law exponents between micropillar-compression and our nanoindentation testing is discussed later based on the analyses of the MD simulation results and a developed dislocation avalanche model. Here, another question arises: why is the obtained exponent for FCC Cu $\sim 6.4$ in our study much higher than the exponent $\sim 1.6$ reported in the nanoindentation experiments and simulation for the FCC metals? The difference is a result of a "first-subsequent mixed analysis" using all of the first and second and subsequent pop-in data analyzed together without separating the first pop-in. To confirm this, we plotted $\Delta h$ for our BCC Fe and FCC Cu data without separating the first pop-in using the logarithmic binning (Supplementary Fig. 24 in Supplementary Note 11), that is, the same method as the papers[19,20] used. Accordingly, we actually estimated the similar power-law exponent $\sim 1.6$ for FCC Cu using our experimental data (see Supplementary Fig. 24c in Supplementary Note 11).

Meanwhile, we would emphasize that especially for the BCC Fe cases, the first–subsequent mixed analysis does not show the power-law scaling at all because a clear separation does exist between the first and the second and subsequent pop-in distributions, as seen in Supplementary Fig. 24a, b. The straightforward reason for the separation is that BCC Fe tends to have a much larger first pop-in magnitude relative to the second and subsequent pop-in magnitudes compared with FCC Cu. This occurs because BCC metals typically have an almost 50% higher ideal shear strain than FCC metals[28], which was defined as a necessary affine shear strain to reach a stress state exhibiting the ideal shear strength (critical shear stress in the dominant slip system)[28,29]. In other words, a perfect crystal maintains its mechanical stability up to the ideal shear strain at which dislocation nucleation can be triggered. Higher ideal shear strain results in a larger first pop-in depth $h$, and thus a larger pop-in load $P^{\text{pop-in}}$. This eventually results in a larger first pop-in magnitude because a larger elastic energy is stored immediately before the first pop-in.

The above-mentioned fact implies that the first pop-in separation is necessary to unveil the unique high-exponent universal-scaling nature hidden in the nanoindentation testing. The statistical independency plots (Supplementary Figs. 22 and 23) also demonstrate the fundamental difference between the first and second and subsequent pop-ins; the pop-ins occurring at $h <$ 40.0 nm (Fe(100)), $h <$ 40.0 nm (Fe(111)), and $h <$ 20.0 nm (Cu (100)) are mostly the first pop-in showing a different trend from

the following pop-ins, which is also confirmed in the plots of the actual load-displacement curves (Supplementary Figs. 35–37). It is worth noting that even at high temperatures, such as at 373 K, high power-law exponent $\beta = 5.0$ was also observed (see Supplementary Note 12), whereas the power-law exponent slightly decreased with the increase in temperature; our dislocation avalanche model suggests the temperature dependency of the power-law exponents (see Supplementary Note 13).

Furthermore, we investigated the distributions of the first and subsequent pop-in magnitudes in the above-mentioned displacement-controlled MD simulations at 5 K on the BCC Fe(100) and Fe(111). In the MD simulations, a force drop[19] and "fictitious" displacement burst were employed for measuring the pop-in magnitude (for definition details of the force drop and fictitious displacement burst and sampling algorithm of these data from the MD load-displacement curve, see Supplementary Note 14). The results are given in Supplementary Note 9. In both the force drop and the fictitious displacement burst plots, we can see the universal power-law scaling distribution in the subsequent pop-ins with high power-law exponents, which agree reasonably well with the experimental power-law exponents; however, large-scale events should be truncated in MD because of the model size limitation.

The high exponents and the materials and surface-orientation dependencies may be attributed to the unique boundary condition and stress and dislocation distributions of nanoindentation, which are different from the uniaxial-loading pillar-compression testing. We recall that the displacement burst results from the motion of dislocation ensembles, and the magnitude of the burst is proportional to the total migration distance of dislocations. The driving force of dislocation (more exactly it is a "dislocation segment"; however, simply, "dislocation" is used hereafter for simplicity) in a material is the total stress exerted on the dislocation, which originates from both the application of an external force to the target material and the stress field produced by other dislocations. The fundamental difference between these testing methods is the force-applying geometry and the resulting stress field.

In micropillar load-controlled compression testing (to a single crystal of pure metal), background stress distribution should be uniform over a slip plane because the exerted stress distribution on a slip plane originated from applying an external force. Hence, once dislocation starts to move, the dislocation motion can only be suppressed and then terminated by dislocation–dislocation interactions. Thus, dislocation multiplication is the major termination mechanism of the displacement burst (dislocation avalanche) in micropillar compression testing[30], while dislocation escape from the free surface is another minor termination mechanism. Note that in extremely small nanopillar testing, the dominant termination mechanism is reversed. In this case, dislocation nucleation from the free surface, and then passing through the entire sample and escaping from the free surface, can be the major mechanism because of the small probability of dislocation multiplication, and thus the small possibility of dislocation–dislocation interaction[31,32].

However, in nanoindentation testing, the background stress distribution produced by the indenter is not uniform[15,33]. Although a very high local background stress field (the level of which is comparable with the ideal shear strength[23,33]) is created in the local region near the indenter tip, the stress level rapidly decreases with an increase in the distance from the indenter tip[23,33]. Thus, the starvation of the driving force owing to the lack of background stress in remote fields far from the indenter tip is one of the reasons for the termination of the displacement burst, in addition to the dislocation–dislocation interactions owing to dislocation multiplication, as we directly show using the MD results as follows.

To observe the dislocation activities and the corresponding stress distribution change and atomic motion during the first and subsequent pop-ins in the displacement-controlled MD (during the fictitious displacement bursts), we visualized the dislocation pattern immediately before and after the pop-ins, together with von Mises stress-invariant distribution, the change in the von Mises atomic-strain invariant[34], and the atomic displacement along the indenter axis (for the first pop-in, see Fig. 3 for BCC Fe (100), Supplementary Fig. 29 for BCC Fe (111), and Supplementary Fig. 30 for FCC Cu (100); for the subsequent pop-ins, see Fig. 4 for BCC Fe (100), Supplementary Fig. 31 for BCC Fe (111), and Supplementary Fig. 32 for FCC Cu (100) (see also movies in Supplementary Movie 1)). Note that the atomic-strain-invariant visualization allows us to directly observe the atoms contributing to the plastic deformation (displacement burst) produced within a pop-in. Moreover, together with dislocation pattern visualizations before and after the pop-in, the history of dislocation motion during the pop-in can also be determined. The atomic displacement along the indenter axis allows us to observe regions that are contributing to the indenter displacement.

At the first pop-in, vast dislocations were nucleated and spread in a fan-like pattern from the indenter tip (Fig. 3a, b). The dislocation behavior has been observed in a nanoindentation experiment using transmission electron microscopy (TEM)[13]; eventually, a local high-density dislocation field was formed in the near field of the indenter tip. At the same time, the stress distribution beneath the indenter immediately contracted and decreased (Fig. 3c, d), and then the indentation load dropped (Fig. 3e) with the generation of a plastic strain (Fig. 3f) and atomic displacement (Fig. 3g) mostly just beneath the indenter tip.

Then, during the subsequent pop-in, the dislocation field expanded out, and simultaneously, new dislocations were formed at the local high-stress field near the indenter tip (Fig. 4a, b). The dislocations reduced the total stress distribution beneath the indenter (Fig. 4c, d). At the same time, some dislocations existing in the remote fields (indicated by a purple arrow in Fig. 4a, b) traveled further because of an additional pushing force, which originated from the near-field dislocation motions (=a reactive force to the backstress force acting on the near-field dislocations from remote field dislocations) (dislocation motion cascade). These dislocation activities generated a certain amount of plastic strain (Fig. 4f) and atomic displacements (Fig. 4g), which contributed to indenter displacement. However, this pushing force does not increase forever because the following dislocations also lose the driving force for these motions when they enter the remote fields with a lower background stress. Moreover, the background stress distribution itself decreases under the constant pop-in load because of a decrease in the contact stress (stress drop) between the indenter and the target material due to the increase in the contact area with the progress of the pop-in. Thus, the dislocations existing in the remote fields will be stopped, as seen in Fig. 4 and in Supplementary Figs. 31 and 32. Meanwhile, the near-field dislocations exhibited a vigorous activity with generating plastic strain to accommodate the indenter tip motion, which was nucleated in the subsequent pop-in at the very local high-stress field beneath the indenter. These dislocations directly interact with each other owing to the high local dislocation density, and most of them seem to become immobile immediately after generating a certain amount of plastic strain and indenter displacement with the reduction in the remote field stress. At this stage, the near-field dislocations can no longer contribute to the generation of further plastic strain and thus the indenter displacement either through its motion or by pushing other dislocations. Note that some of the near-field dislocations (indicated by the orange arrow in Fig. 4a, b) escaped out to the remote fields

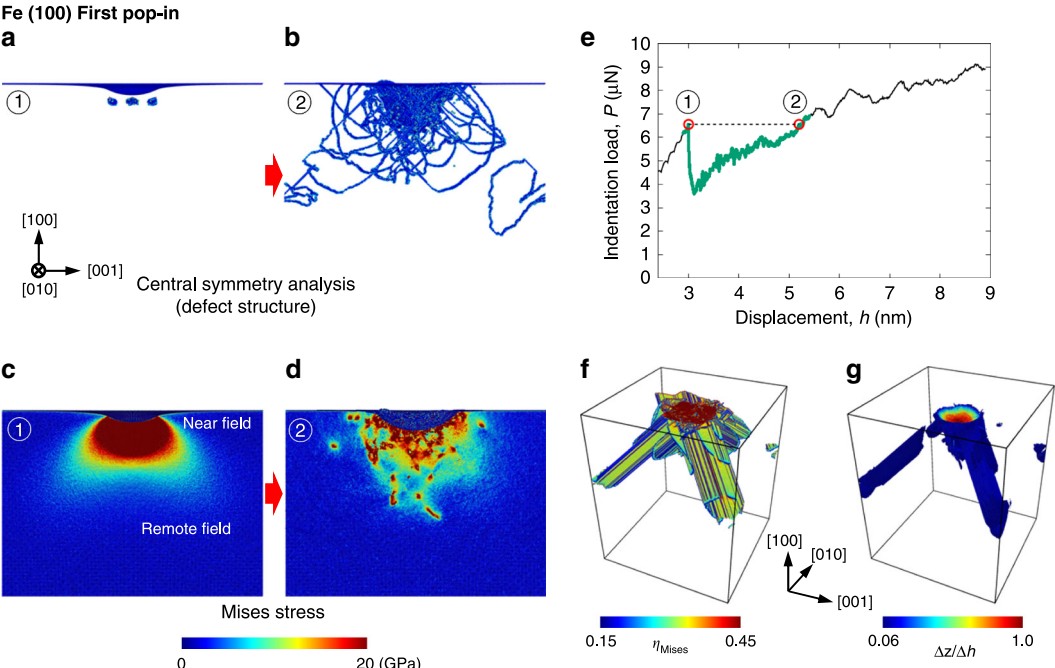

**Fig. 3 First pop-in behavior in molecular dynamics simulation at 5 K on the (100) BCC Fe surface.** Defect structure immediately (**a**) before and (**b**) after the first pop-in (central symmetry parameter coloring[43]), von Mises stress distribution immediately (**c**) before and (**d**) after the first pop-in, and (**e**) the corresponding load-displacement curve. **f** Spatial distribution of von Mises atomic-strain-invariant $\eta_{Mises}$. Only atoms satisfying $\eta_{Mises} > 0.15$ are displayed. **g** Spatial distribution of atomic displacement along loading direction ([100]), $\Delta z$, normalized by displacement burst $\Delta h$. Only atoms satisfying $\Delta z/\Delta h > 0.06$ are shown. The von Mises stress distribution is shown on a (010) plane passing through the indenter central axis. The movies of the dislocation pattern and the stress-distribution evolution during the simulation can be found in Supplementary Movie 1. See also Supplementary Note 15 for (111) BCC Fe and (100) FCC Cu.

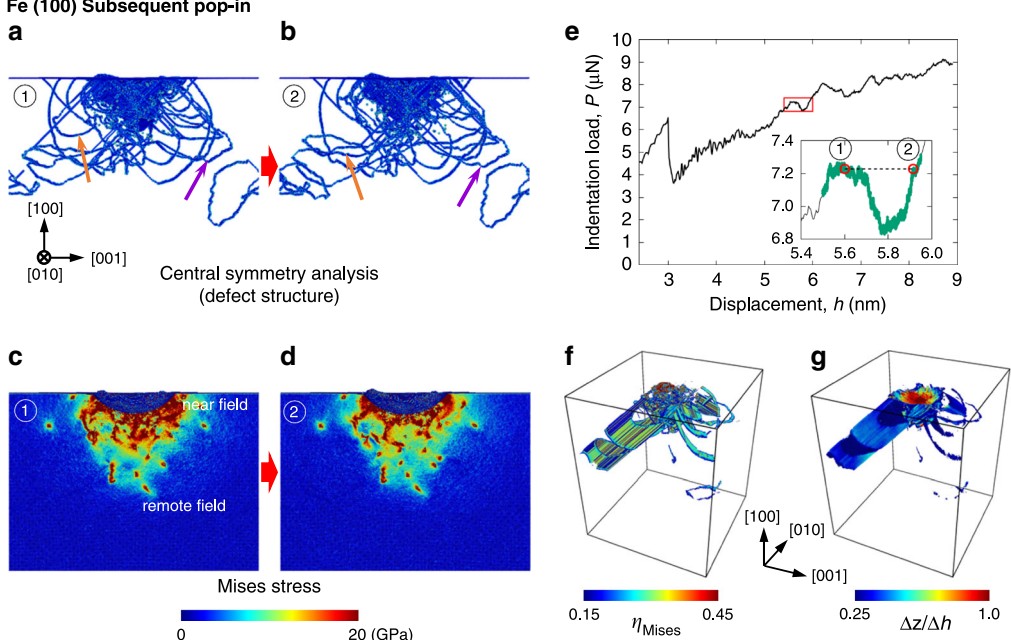

**Fig. 4 Subsequent pop-in behavior in molecular dynamics simulation at 5 K on the (100) BCC Fe surface.** Defect structure immediately (**a**) before and (**b**) after a subsequent pop-in (central symmetry parameter coloring[43]), von Mises stress distribution immediately (**c**) before and (**d**) after a subsequent pop-in, and (**e**) the corresponding load-displacement curve. **f** Spatial distribution of von Mises atomic-strain-invariant $\eta_{Mises}$. Only atoms satisfying $\eta_{Mises} > 0.15$ are displayed. **g** Spatial distribution of atomic displacement along loading direction ([100]), $\Delta z$, normalized by displacement burst $\Delta h$. Only atoms satisfying $\Delta z/\Delta h > 0.25$ are displayed. The von Mises stress distribution is shown on a (010) plane passing through the indenter central axis. The movies of the dislocation pattern and the stress-distribution evolution during the simulation can be found in Supplementary Movie 1. See also Supplementary Note 16 for (111) BCC Fe and (100) FCC Cu.

with the generation of a certain indenter displacement, but these were also stopped eventually when the backstress from the remote field dislocations, pushing force from the near-field dislocations, and the background stress were balanced.

All the above-mentioned unique termination mechanisms of the dislocation motion in nanoindentation testing originate from the nonuniform stress and dislocation distributions with a rapid decay with respect to the distance from the indenter tip, in addition to the dislocation–dislocation direct interaction in the near field. Hence, the pop-in owing to the dislocation avalanche is fundamentally restricted by the unique stress and dislocation distributions of nanoindentation testing by restricting the dislocation motion, which is not formed in the micropillar-compression testing. The additional restriction is the reason for high power-law exponents, i.e., the probability of large-scale events becomes significantly small.

On the basis of the above-mentioned discussion and MD observations concerning the dislocation activities in the unique nanoindentation stress field, we would propose a dislocation avalanche model in the unique nanoindentation stress field (see Supplementary Note 13). The model can successfully explain the origin of the power-law exponents in the second and subsequent pop-ins, which is related to the materials' intrinsic properties, temperature, and the surface property; however, further studies are necessary for the quantitative determination of the power-law exponents. Moreover, the model can also explain the reason for the difference in the power-law exponent between the micropillar compression and nanoindentation testing.

In summary, the distribution of pop-in magnitude transitions from Gaussian-like for the first pop-in to power-law-like for the second and subsequent pop-ins, as demonstrated by nanoindentation testing. The Gaussian-like distribution of the first pop-in was consistent with the theoretical distribution based on the thermal activation theory of dislocation nucleation. Thus, the data indicate that the first pop-in is dominated by a thermal activation process of dislocation nucleation, as has been reported in many past studies. More importantly, the second and subsequent pop-ins are dominated by dislocation avalanches, which follow power-law statistics. The power-law exponent for this distribution was much larger than that for uniaxial pillar-compression testing. The difference may be attributed to the spatial inhomogeneities in the stress and dislocation density. Thus, it can be concluded that inhomogeneous mechanical deformation belongs to a different universal-scaling class than uniaxial deformation.

## Methods

**Nanoindentation experiment**. The (100) and (111) surfaces of pure BCC Fe and the (100) surface of pure FCC Cu single crystals were electropolished before the experiments were conducted. The investigation of the oxide layer forming on the surface of the BCC Fe is discussed in Supplementary Note 17. The indents were made by a Hysitron TI950 instrument with a Berkovich indenter (Bruker Co.) under the load-controlled mode. The nanoindentation experiments were conducted at room temperature (300 K). The loading and unloading rates of the indenter were 50 $\mu Ns^{-1}$ with a holding segment of 10 s at a peak load of 1 mN. The indenter tips in these nanoindentation tests for the (100) and (111) surfaces in BCC Fe, and the (100) surface in FCC Cu had a radius of $R = 391$, 365, and 752 nm, respectively, which were calculated from a loading curve in the initial elastic region below the first pop-in based on Hertz's contact theory by substituting a reduced modulus that was independently measured from the unloading curve.

A typical atomic force microscope (AFM) image of the (100) sample surface, including an indent, is shown in Supplementary Note 18. The average roughness ($R_a$), maximum peak-to-valley roughness ($R_{max}$), and average wavelength of the profile ($\lambda_a$) of the sample surface are 0.303, 2.18, and 355 nm, respectively, based on a line-profile analysis with 10μm length. The indents were made at random positions in five different regions of $200 \times 100$ μm$^2$ in a size within the polished area of the sample surface, which was 3 mm in diameter. Two-hundred indents were made in each region with a pitch of 10 μm, which was large enough to avoid interactions between the indent marks; thus, 1000 indentations were made on each sample. All the indentation load-displacement curves are shown in Supplementary Note 19.

Concerning the quadratic area function $A(h) = A(h_{con}(h)) = c_2(h_{con}(h))^2 + c_1 h_{con}(h) + c_{\frac{1}{2}}(h_{con}(h))^{\frac{1}{2}}$ ($c_2 = 24.5$, $c_1 = 2.61 \times 10^3$ nm, and $c_{\frac{1}{2}} = 1.57 \times 10^{-7}$ nm$^{\frac{3}{2}}$) considering the effect of the rounded indenter tip shape using the Oliver–Pharr (OP) method[22], a contact depth $h_{con}(h)$ is given as $h_{con}(h) = h - P(h)/S(h)$, $P(h)$ is the measured load, $S(h) = (2/\sqrt{\pi})E_r\sqrt{A(h_{con}(h))}$ is the effective stiffness, and $E_r$ is the reduced elastic modulus defined using the Young's modulus and Poisson's ratio of both the target and the indenter materials. The reduced elastic modulus $E_r$ was obtained from $S(h) = (2/\sqrt{\pi})E_r\sqrt{A(h_{con}(h))}$ using directly measured $S(h)$ as a slope of the load-displacement curve immediately after the unloading starts at 1mN loading. We measured 200 $E_r$s at five independent regions and obtained an average in each region. The averages dispersed only within 1.5% with the total averages of 198, 197, and 117 GPa for (100) in BCC Fe, (111) in BCC Fe, and (100) in FCC Cu, respectively, which demonstrates that there were no significant regional variations. Therefore, these averages were used for the contact-area analysis. Here, each average was calculated from an average of 1000 unloading slopes of load-displacement curves in each region. The contact area $A(h)$ was obtained by numerically solving these equations. The area function was preliminarily calibrated for our indenter tip using a standard sample of fused silica by the Oliver–Pharr method[22] (see also Supplementary Note 3).

**Nanoindentation MD simulation**. Atomistic slab models of BCC Fe with orientations—$x$: [001], $y$: [010], $z$: [100] ((100) surface model) and $x$: [1$\bar{1}$0], $y$: [11$\bar{2}$], $z$: [111] ((111) surface model)—were constructed. The dimensions of the models were 71.1 nm × 71.1 nm × 71.1 nm for the (100) surface model and 71.1 nm × 71.3 nm × 71.2 nm for the (111) surface model. The numbers of atoms were 30,876,498 for the (100) surface model and 31,021,056 for the (111) surface model. The embedded atom method (EAM) potential for Fe[35] was used to describe the interatomic interactions. The lattice constant and elastic constants were estimated as $b = 2.855$ Å, $C_{11} = 243$, $C_{12} = 145$, and $C_{44} = 116$ GPa, which agree with the experimentally determined values of $b = 2.860$ Å[36], $C_{11} = 243$, $C_{12} = 138$, and $C_{44} = 122$ GPa[37].

An atomistic slab model of FCC Cu with an orientation $x$: [001], $y$: [010], $z$: [100] ((100) surface model) was constructed. The dimension of the model was 70.5 nm × 70.5 nm × 76.0 nm. The number of atoms was 31,941,000. The embedded atom method (EAM) potential for Cu[38] was used to describe the interatomic interactions. The lattice and elastic constants were estimated as $b = 3.615$ Å, $C_{11} = 179$, $C_{12} = 123$, and $C_{44} = 81.0$ GPa, which agree with the experimentally determined values of $b = 3.615$ Å[39], $C_{11} = 170$, $C_{12} = 123$, and $C_{44} = 75.8$ GPa[40].

Before starting indentation simulations, the models were first equilibrated using the Parrinello–Rahman NPT ensemble method[41,42] for 50 ps at an in-plane normal stress of 0 Pa at simulation temperatures of 5 K (for (100) and (111) surfaces of BCC Fe and (100) surface of FCC Cu) and 500 K (for (100) and (111) surfaces of BCC Fe) to release the in-plane stresses. The $z$ position of the spherical indenter with radius $R_{sim} = 15$ nm was controlled to move it along an axis perpendicular to the model surface with 5 ms$^{-1}$. During the simulations, the center of mass of the atomic slab model was fixed, and the $x$ and $y$ dimensions of the slab model were relaxed such that the normal stress was 0 Pa in these directions. The following repulsive force was assumed to act between the indenter and the slab model: $F(r) = -K(r - R_{sim})^2$; $r < r_c$, where $r$ denotes the distance of the atoms in the target material to the centroid of the spherical indenter tip, $K$ denotes a force constant, which was set to 10 eV Å$^{-3}$, and $r_c$ denotes the potential cutoff distance, which was set to 0.53 nm for Fe and 0.55 nm for Cu, respectively.

The von Mises atomic-strain invariant was computed on the basis of the atomic displacement of each atom from the beginning to the end of the pop-in with a cutoff strain of 0.1 to avoid the detection of change in elastic strain, which typically must be <0.1.

It should be noted why the displacement-controlled mode was used in the MD simulation instead of the dynamic load-controlled mode as in our experiments. In the load-controlled mode, a dynamic motion of the indenter tip during the pop-in should be solved in real time. However, dynamic parameters, such as the apparent inertia mass of the indenter tip and the effective damping factor of the nanoindentation system, are unfortunately unknown. Though the dynamic motion could be computed with appropriate dynamic parameters, the timescale in MD must be too short for direct comparison with the experiments. Because such unknown parameters are not necessary in the displacement-controlled mode, we decided to use the displacement-controlled mode, while the time-scale issue still exists, and the displacement burst cannot be directly obtained. Because the size of the indenter tip in the MD simulation was smaller than that in the actual nanoindentation testing owing to the limitation of computational resources, the volume of the plastic zone generated beneath the indenter was also smaller. However, the essential features of the dislocation activities, such as dislocation nucleation in the first pop-in and dislocation network evolution in the subsequent pop-ins, can be qualitatively demonstrated in the MD simulation.

## Data availability

The data that support the findings of the study are available from the corresponding authors upon reasonable request.

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

## Acknowledgements
The authors thank Ms. Eri Nakagawa for supporting the experiments. Y.S. acknowledges the support by JSPS KAKENHI Grant No. JP19J10309. S.S. acknowledges the support by JSPS KAKENHI Grant No. JP19K23487. T.O. acknowledges the support by JSPS KAKENHI Grant Nos. JP16H06366 and JP18H01696. T.H. acknowledges the support by JSPS KAKENHI Grant No. JP16H06478. S.O. acknowledges the support by JSPS KAKENHI Grant Nos. JP18H05453, JP17H01238, and JP17K18827. T.O. and S.O. acknowledge the support by Element Strategy Initiative for Structural Materials (ESISM) of MEXT, Grant No. JPMXP0112101000.

## Author contributions
T.O. and S.O. designed the project. T.O. conducted the experimental work. Y.S. and S.S. conducted MD simulations. Y.S., T.H., and S.O. developed the theory. S.O. guided MD simulations and theoretical analysis. Y.S. and S.O. wrote the paper. All authors contributed to the discussion of the results.

## Competing interests
The authors declare no competing interests.
