## [Peer Review File · Nature Communications]

Reviewers' Comments:

Reviewer #1:

Remarks to the Author:

In this manuscript, the authors investigated experimentally the statistical effects hidden in the noise of force-depth nanoindentation curves in BCC iron. The authors find that the distribution of the pop-in magnitudes transitions from Gaussian-like for the first pop-in to power-law-like for the second and subsequent pop-ins, and they demonstrated it by nanoindentation testing with measurements of the contact stress drop. The Gaussian-like distribution of the first pop-in appears consistent with the theoretical distribution based on the thermal activation theory of dislocation nucleation. Thus, the authors' data indicate that the first pop-in is dominated by a thermal activation process of dislocation nucleation, as reported in many past studies. Further, the authors conclude that more importantly, the second and subsequent pop-ins are dominated by dislocation avalanches, which follow power-law statistics.

This is a well written manuscript with an important and novel conclusion that should become publishable in a high impact journal such as Nature Communications. However, there are several steps to be performed before recommending such a publication. My concerns and questions are posed below:

1. The authors state that power-law exponents have not been identified in experimental testing of nanoindentation. This is not true since in:
Crystals 2019, 9(12), 652; <https://doi.org/10.3390/cryst9120652>
there is a clear study of various FCC metals in a very high-throughput study of nanoindentation force-depth curves. There, first pop-ins were not separated from second pop-ins, but other conditions (Berkovich tip, load-control) appear similar. There, it appears that the exponent the authors investigated is closer to 1.6-2.0 for FCC metals. This is a very interesting difference that not only should be mentioned by the authors in their manuscript, but also should be investigated if possible (by testing an FCC metal, such as Ni), in order to strengthen significantly their message.
2. The authors display a Force-Depth curve in Fig.1 that suggests that their maximum depth is approximately 100nm. This statement is also supported by their description in Methods, where they emphasize that they set their max force to 1mN (as shown in Fig.1). However, the Supplementary Information contains a post-indent image (last page) that seems to suggest that the indents went much deeper than 100nm. Is it possible to include (possibly in the supplement) a figure with all indentation curves that were utilized for producing the distributions in Fig.2 ?
3. The authors fit the statistical distribution of the first pop-ins to a rate nucleation model (Eq.2). While this is very interesting, there is a crucial dependence on temperature that needs to be confirmed in some way. To my understanding, no prior publication has explicitly demonstrated a statistical dependence of these events on temperature, and the authors' model is crucially dependent on it. It is imperative that the authors demonstrate in some way that the experimental data of the first pop-ins contain significant temperature dependence that could be modeled in such a simple manner. While performing an appropriate temperature-dependent experiment would be ideal, another possibility would be that the authors find data from other publications on the subject that confirm this model.
4. The first pop-in distributions in Fig.2 appear skewed in log scale. Is it possible that the distributions are also shown in linear scale?
5. The molecular dynamics simulations in Fig.3 are interesting. However, it is not clear to me why the authors chose to not show the event distributions from these simulations? Don't they have enough events? It would be important that the authors comment on this issue, since MD simulations can provide insights for such statistical effects, and especially the temperature dependence of the first pop-in events.

6. Another issue with Fig.3 and the associated discussion in the main text is that the event definition is confusing. For a displacement-controlled simulation, one should define "equivalent" displacement bursts out of the force-drops, something that is discussed in detail in Ref.[19] for example. The definition of a displacement burst out of the displacements that are "recovering" the load, should be at least identified as a "specially" defined one, and should be mentioned as one. The issue with this definition is that the aforementioned displacements are force-balanced by the applied loads, so they are not formally avalanche events, which should be by definition out-of-equilibrium. That said, the authors can only choose to focus on the force drops, which can be associated to displacements through appropriate unit change by using the elastic modulus.

7. In the MD simulations, is there any significant difference between the statistical distributions of the first event, compared to the next ones? Can the authors comment on any evidence they may collect?

Reviewer #2:

Remarks to the Author:

The manuscript presents an interesting study of the stress and strain burst statistics during nano-indentation of iron single crystals. The authors present experimental results and MD simulations showing that pop-ins are power law distributed, apart from the first nucleation event that is approximately Gaussian distributed. While I judge that the work has merit and could be accepted in Nature Communications, the authors did not exploit fully the results they have at hand and some obvious questions remain unanswered. I suggest a series of additional analysis that should strengthen the conclusions of the paper.

-The authors claim that displacement steps Δh are not power law distributed, while stress drops are. I can not see any convincing evidence for this. The "power laws" in Fig. S1 are as good (or as bad) as those in Fig. 2. I suggest to bin in logarithmic scale: this is conventionally done for power law distributions. Please also show bin-free distributions by reporting (in the supplement) the cumulative distribution by simply rank-ordering the recorded values of Δh (and $\Delta\sigma$).

-The authors make a complicated argument with several assumptions to measure stress drops. While all this might be reasonable, I would like to see a demonstration. This could be obtained by MD simulations where it is possible to have at the same time access to stresses and displacements. The authors should verify that their relation between stress drops and displacement is correct.

- It is argued that MD simulations can not be compared with experiments because of the different loading mode. The argument is not very convincing. I do not understand the claim that displacement control was needed "to maintain stability". One can certainly apply a force to the indenter and slowly increase it and I can not see how this will affect the stability of the simulation. But even if there is a practical reason for which force driven simulations are unfeasible (and the authors should explain it with much more care), the authors should still do more to compare experiments and simulations: for instance they should measure the pop-in stress drop distribution from MD simulations. It is really a pity that almost no information is extracted from the MD simulations.

- Are the Δh steps statistically independent on time? Clearly the first step is larger but are all the other steps statistically equivalent? To test this the authors could plot $\langle \Delta h | h \rangle$ vs h (i.e. average step size at given displacement level) and/or $\langle \Delta h | k \rangle$ with k being the pop-in number (first, second, etc). Since $N=1000$ indentations are performed this should not be a problem. The same analysis should be performed for the stress drop. Doing this is extremely important to understand the universality class of the distribution. If the Δh (or $\Delta\sigma$) signal is not

stationary then the high exponent of the distribution (i.e. $\beta=2.8$) can be explained as a result of the integration of a non-stationary signal. This is discussed in previous papers on crackling noise, see for instance Durin, G., & Zapperi, S. (2006). The role of stationarity in magnetic crackling noise. *Journal of Statistical Mechanics: Theory and Experiment*, 2006(01), P01002.

- minor point: I do not understand the role of c in the x-axis of Fig. 2. The c constant is not discussed anywhere and looks completely useless (since of course c cancels out!).

Reviewer #3:

Remarks to the Author:

Manuscript ID: NCOMMS-19-41235

Title of Article: Unique universal scaling in nanoindentation pop-ins

This manuscript describes a series of nanoindentation experiments performed on iron single crystals, and preliminary analysis of molecular dynamics simulations of the same process. The central claim is that the contact stress drop of the initial pop-in roughly follows a Gaussian distribution, whereas the contact stress drop of the subsequent pop-ins roughly follows a power law distribution. The exponent of the power law is empirically measured to be higher than those generally observed in micro-pillar plasticity. This is suggested as evidence that the phenomena belong to different universal scaling classes.

Before continuing with the techniques and analysis, I would like to discuss the significance of the claims. Whether or not the manuscript provides sufficient support for the claim that the physical process underlying the second and subsequent pop-ins belongs to a different universality class than that behind micro-pillar plasticity, it is unclear why this would be a result of general interest. I would certainly be interested in a precise explanation of the differences between these processes, particularly with regard to dislocation interaction and multiplication. That said, just reporting that they occur with different probability distributions of contact stresses does not substantially increase my understanding. As for the question of universality class, the knowledge that two processes belong to the same universality class (in principle) allows the transfer of some understanding about one process to the other. That is not done here, with the claim of differing universality classes amounting to the tautological statement "different processes are different".

I do wish to acknowledge that the experiments are not trivial to perform, and that the authors have collected a substantial quantity of data about a question that could be of general interest. The analysis and conclusions feel incomplete though, and partly for that reason I do not believe that this manuscript would substantially change thinking in the field.

I have several concerns with the techniques and analysis. These are enumerated below:

1. The definition of a second pop-in is not precise. The authors set an arbitrary velocity threshold above which the event is identified as a pop-in, but this is unsatisfactory for several reasons. First, have the authors verified that the events they sample are not simply electrical and mechanical noise? The noise in electrical instruments is often pink noise, which could explain the observed power law probability distribution they observe in Figure 2. One natural resolution would be to consider all events after the first pop-in, and to see whether there is a natural decomposition of the resulting probability distribution into a noise contribution for the smallest events and everything else. Second, the main characteristic of the first pop-in is that there is irreversible plastic deformation during the event, but not before. The unloading curves in Figure 1 indicate that there is irreversible plastic deformation all along the loading curve after the first pop-in. Then are the points identified as subsequent pop-ins at all distinguishable from the rest of the loading curve on physical grounds?

2. Perhaps the difficulty with identifying discrete subsequent pop-ins is that there is effectively continuous thermal activation of dislocation-related events. This could be suppressed by increasing the indentation rate and decreasing the temperature, thereby strengthening the claim that the subsequent pop-ins really are significant events. This could also have the effect of allowing the subsequent pop-ins to be more directly related to the athermal activation energies, allowing a more fundamental characterization of the phenomenon. Where is the temperature reported?
3. Given the importance of the contact area to the calculation of the stress drop magnitudes, it is surprising and concerning that the authors did not verify the assumption that the contact area is proportional to the square of the indentation depth. Such verification would be relatively straightforward, involving a sequence of interrupted tests and subsequent investigation of the surface profile.
4. The authors do not appear to address the possible formation of an oxide layer on the surface of the Fe. Such an oxide layer could form very quickly indeed, and be related to the difference between the first pop-in (film rupture) and subsequent pop-ins (dislocation multiplication). What assurance can the authors provide that such an oxide layer is not present?
5. Have the authors considered fitting the probability distribution of initial pop-ins using the well-established model of Schuh and Lund (DOI: 10.1557/JMR.2004.0276)? What is the advantage of the model derived in the supplementary material over that one?
6. The simulations do not provide any atomistic insight into the processes occurring during the so-called subsequent pop-ins, and have been performed many times before in the literature. It is unclear what value the simulations add to the manuscript.
7. Why does the power-law exponent reported here differ so substantially from that reported in Ref. 19? That the power-law exponent in Ref. 19 is entirely consistent with the exponent for micro-pillar plasticity suggests that this manuscript is the outlier.
8. What is the algorithm that the authors use for the fitting of the models? I don't see any mention of this, or the resulting uncertainties in the fitting parameters.

Responses to Reviewer 1

“In this manuscript, the authors investigated experimentally the statistical effects hidden in the noise of force-depth nanoindentation curves in BCC iron. The authors find that the distribution of the pop-in magnitudes transitions from Gaussian-like for the first pop-in to power-law-like for the second and subsequent pop-ins, and they demonstrated it by nanoindentation testing with measurements of the contact stress drop. The Gaussian-like distribution of the first pop-in appears consistent with the theoretical distribution based on the thermal activation theory of dislocation nucleation. Thus, the authors' data indicate that the first pop-in is dominated by a thermal activation process of dislocation nucleation, as reported in many past studies. Further, the authors conclude that more importantly, the second and subsequent pop-ins are dominated by dislocation avalanches, which follow power-law statistics.

This is a well written manuscript with an important and novel conclusion that should become publishable in a high impact journal such as Nature Communications.”

We are very pleased that the reviewer thinks that the paper is well written and publishable in Nature Communications. We truly appreciate the reviewer's valuable comments. We have carefully studied the reviewer's comments and revised our manuscript accordingly.

Before addressing the reviewer's comments, we would like to discuss revisions to the statistical analysis method for subsequent pop-ins. In the revised manuscript, we simply employed the displacement burst Δh , which is directly measured in indentation testing as the measure of the pop-in magnitude for the subsequent pop-in analysis instead of the stress drop $\Delta\sigma$, which has been previously used in the manuscript; we still use the stress drop for the first pop-in analysis. We made these changes because we received two valuable comments from Reviewer #2: 1) “ Δh seems to also follow the power law” and 2) “statistical independence with respect to the indentation depth h of the subsequent pop-in data should be checked.” To confirm comment 2), we plotted $\langle \Delta h | h \rangle$ vs. h and $\langle \Delta\sigma | h \rangle$ vs. h , i.e., conditional probabilities; an average displacement burst Δh and an average contact stress drop $\Delta\sigma$ at a given indentation depth h are shown in Figs. S10-1 and S10-2, respectively. It is clear that Δh exhibits a nice statistical independence within a wide range of indentation depth, i.e., $20.0\text{--}40.0\text{ nm} \leq h \leq 90.0\text{--}145.0\text{ nm}$, while $\Delta\sigma$ shows a weak depth dependence, as seen in Fig. S10-2. Physically, the displacement burst Δh is directly related to the external work done by an indenter under the assumption that the constant indentation load is maintained over the pop-in event, while the previously used stress drop $\Delta\sigma$ is based on the force balanced states before and after the pop-in. Therefore, we decided to use the Δh data for the

subsequent pop-in power-law analyses within the statistically independent displacement range even though $\Delta\sigma$ can acceptably demonstrate the power law, as seen in Fig. S4-1 in Supplementary information S4. The newly estimated power law exponents using Δh are $\beta = 5.6$ for the BCC Fe (100) surface and $\beta = 3.9$ for the BCC Fe (111) surface, as shown in revised Fig. 2. Here, according to the comment from Reviewer #2 for data analysis, the exponents were calculated from cumulative distributions, as shown in revised Fig. 2. For reference, equal-width binning and logarithmic binning plots are also shown in Fig. S5-1. Moreover, in addition to BCC Fe, according to Reviewer #1's valuable comment, we also performed nanoindentation testing for the FCC Cu (100) surface and obtained $\beta = 6.4$. Nevertheless, all power law exponents are still significantly higher than those obtained by the usual pillar-compression testing ($\beta = 1.0$ – 1.8) and by nanoindentation testing and simulations, $\beta \cong 1.6$ (Reference [20] (R. Bolin et al., Crystal 2019) and Reference [19] cited in our manuscript). Of note, for the first pop-in analysis, we also used Δh in addition to $\Delta\sigma$ to perform a "first-subsequent (all-data) mixed analysis," as discussed later. Moreover, we developed a novel dislocation avalanche model (please see Supplementary information S13). The model can successfully explain the origin of power-law exponents in the second and subsequent pop-ins, which is related to material properties and surface orientation, and can explain also the reason for the difference in the power law exponent between the micro-pillar compression and nanoindentation testing.

"However, there are several steps to be performed before recommending such a publication. My concerns and questions are posed below:

1. The authors state that power-law exponents have not been identified in experimental testing of nanoindentation. This is not true since in:

Crystals 2019, 9(12), 652; <https://doi.org/10.3390/cryst9120652>

there is a clear study of various FCC metals in a very high-throughput study of nanoindentation force-depth curves. There, first pop-ins were not separated from second pop-ins, but other conditions (Berkovich tip, load-control) appear similar. There, it appears that the exponent the authors investigated is closer to 1.6-2.0 for FCC metals. This is a very interesting difference that not only should be mentioned by the authors in their manuscript, but also should be investigated if possible (by testing an FCC metal, such as Ni), in order to strengthen significantly their message."

We regret not referencing such an important paper directly related to our study, which addressed the statistical analysis of nanoindentation pop-ins for FCC metals in the same way, while the first pop-in was not separated in the statistical power law analysis. We have added the paper in the references

and introduced the paper in the introduction.

Upon the reviewer's advice, we also investigated the FCC Cu (100) surface in addition to BCC Fe (100) and (111) surfaces. Furthermore, the FCC Cu results exhibit a Gaussian-like distribution in the first pop-in and a power law in the subsequent pop-ins and Fe results, as seen in the revised Fig. 2 in the manuscript. The obtained power law exponent of the subsequent pop-ins (red open triangle plots) was estimated as $\beta = 6.4$ from its cumulative distributions, which is much higher than that of usual pillar-compression testing and even nanoindentation testing. Here, the question arises why our exponent $\beta = 6.4$ for FCC Cu is much higher than exponents $\beta < 2.0$ (~ 1.6) for FCC metals, which have been reported in nanoindentation experiments and simulations (Reference [20] (R. Bolin et al., *Crystal*, 9, 1 (2019)) and Reference [19] (H. Song et al., *J. Mech. Phys. Solids*, 123, 332 (2019)) cited in our manuscript). After the following additional analysis "without" separating the first pop-in data [first-subsequent (all-data) mixed analysis], we strongly believe that the difference is a result of whether the first pop-in is separated from other pop-ins. To demonstrate this, we re-plotted Δh for both Fe and Cu, as shown in Fig. S11-1 (in Supplementary information S11), "without" separating the first pop-in data and using the same logarithmic binning analysis as that used in the nanoindentation paper (R. Bolin et al., *Crystal*, 9, 1 (2019)) for the same FCC Cu. Actually, we obtained a similar power law exponent $\beta \cong 1.6$ as the one that has been reported in the past nanoindentation papers [please see Fig. S11-1(c)]. In addition, we also emphasize that especially for BCC Fe [Fig. S11-1(a)(b)], the "first-subsequent mixed analysis" does not show power law scaling because a clear separation exists between the magnitude distributions of the first and subsequent pop-ins, as seen in Fig. S11-1(a)(b). Moreover, the statistical independence analysis of Δh with respect to indentation depth h (please see Supplementary information S10) suggests that the first pop-in must be different from the others. Hence, the additional analyses for FCC Cu and the "first-subsequent mixed analysis," which have been suggested by reviewer #1, allow us to make the following conclusions: 1) the separation of the first several pop-ins from other pop-ins is necessary and 2) the separation unveils a hidden high-exponent universal-scaling-world in nanoindentation testing.

We have added the above discussions and figures to the manuscript and Supplementary information.

"2. The authors display a Force-Depth curve in Fig.1 that suggests that their maximum depth is approximately 100nm. This statement is also supported by their description in Methods, where they emphasize that they set their max force to 1mN (as shown in Fig.1). However, the Supplementary Information contains a post-indent image (last page) that seems to suggest that the indents went much deeper than 100nm. Is it possible to include (possibly in the supplement) a figure with all

indentation curves that were utilized for producing the distributions in Fig.2?”

We have changed the figure of the post-indent image to an image with the typical indentation depth of ~100 nm (Fig. S18-1). In addition, according to the reviewer’s suggestion, we have added figures that show all indentation load-displacement curves (Figs. S19-1-3) in Supplemental information S19, which were used to obtaining the displacement burst data.

“3. The authors fit the statistical distribution of the first pop-ins to a rate nucleation model (Eq.2). While this is very interesting, there is a crucial dependence on temperature that needs to be confirmed in some way. To my understanding, no prior publication has explicitly demonstrated a statistical dependence of these events on temperature, and the authors' model is crucially dependent on it. It is imperative that the authors demonstrate in some way that the experimental data of the first pop-ins contain significant temperature dependence that could be modeled in such a simple manner. While performing an appropriate temperature-dependent experiment would be ideal, another possibility would be that the authors find data from other publications on the subject that confirm this model.”

Thank you for the valuable comment. We completely agree with the reviewer. Additional nanoindentation testing to confirm the temperature dependence can strongly support our theory for the first pop-in. According to the comment, we performed additional experiments and MD simulations at different temperatures. In particular, we performed 1) nanoindentation experiments at 373 and 473 K on the (100) surface of BCC Fe (80 and 180 K higher than that in other previous indentation tests at 300 K); the results and technical details are given in Supplementary information S8, and 2) molecular dynamics (MD) simulations at both 5 and 500 K on (100) surface of BCC Fe (60 independent MD nanoindentation tests for each temperature and obtain the same number of first pop-in data); the results are given in Supplementary information S9. The technical details of force-drop ΔP and displacement-burst Δh sampling from the load-displacement curve obtained by displacement-controlled MD is described in Supplementary information S14. These experimental and MD results clearly demonstrate the Gaussian-like distribution of first pop-in and its temperature dependence predicted by the theory (Eq. (2)).

We have added the above discussions and figures to the manuscript and Supplementary information.

“4. The first pop-in distributions in Fig.2 appear skewed in log scale. Is it possible that the

distributions are also shown in linear scale?"

According to the reviewer's comment, we have also shown the distribution on a linear scale in Supplementary information S7.

"5. The molecular dynamics simulations in Fig.3 are interesting. However, it is not clear to me why the authors chose to not show the event distributions from these simulations? Don't they have enough events? It would be important that the authors comment on this issue, since MD simulations can provide insights for such statistical effects, and especially the temperature dependence of the first pop-in events."

Thank you for the valuable comments. As we have already discussed above, according to the reviewer's comment, we performed 60 independent displacement-controlled MD simulations for 1) BCC Fe (100) at 5 and 500 K to obtain the event distribution for the first pop-in, its temperature dependence, and the power-law exponent of the second and subsequent pop-ins at 5 K and 2) BCC Fe (111) at 5 K to obtain the power-law exponent of the second and subsequent pop-ins at 5 K; the results are given in Supplementary information S9. Then, the agreements of the first pop-in statistics with the theory and of the second and subsequent pop-in power-law statistics with experiments were demonstrated. Moreover, we have added MD snapshots and movies for dislocation and plastic strain nucleation and evolution and stress field and atomic displacement evolution beneath the indenter during the first and subsequent pop-ins to Figs. 3 and 4 (in the main text) and to Supplementary information S15 and S16 with movies "Movies.pptx."

"6. Another issue with Fig.3 and the associated discussion in the main text is that the event definition is confusing. For a displacement-controlled simulation, one should define "equivalent" displacement bursts out of the force-drops, something that is discussed in detail in Ref.[19] for example. The definition of a displacement burst out of the displacements that are "recovering" the load, should be at least identified as a "specially" defined one, and should be mentioned as one. The issue with this definition is that the aforementioned displacements are force-balanced by the applied loads, so they are not formally avalanche events, which should be by definition out-of-equilibrium. That said, the authors can only choose to focus on the force drops, which can be associated to displacements through appropriate unit change by using the elastic modulus."

Thank you for pointing out this very important issue. We agree with the comment that we should

focus on the force drops. We estimated the indentation force drop in the MD simulations for Fe (100) and (111) at 5 K for the first and subsequent pop-ins and for Fe (100) at 500 K for the first pop-in; again the probability distributions are shown in Supplementary information S9. For reference, we have also plotted the distribution with a “fictitious” displacement burst, which is also shown in Supplementary information S9. The technical details of the force drop estimation from the MD load-displacement curve are explained in Supplementary information S14. In both plots, we observe a Gaussian-like distribution in the first pop-in and a universal power law scaling distribution in the subsequent pop-ins with exponents $\beta = 5.0$ for Fe (100) and $\beta = 3.8$ for Fe (111) using the force drops. The exponents reasonably agree with experimental exponents. For reference, all MD load-displacement curves are shown in Supplementary information S9.

“7. In the MD simulations, is there any significant difference between the statistical distributions of the first event, compared to the next ones? Can the authors comment on any evidence they may collect?”

Thank you for the comment. We have already discussed this point in the reply to comment 1.

Responses to Reviewer 2

“The manuscript presents an interesting study of the stress and strain burst statistics during nano-indentation of iron single crystals. The authors present experimental results and MD simulations showing that pop-ins are power law distributed, apart from the first nucleation event that is approximately Gaussian distributed. While I judge that the work has merit and could be accepted in Nature Communications, the authors did not exploit fully the results they have at hand and some obvious questions remain unanswered. I suggest a series of additional analysis that should strengthen the conclusions of the paper.”

We are very pleased that the reviewer thinks that our work has merit and can be accepted in Nature Communications. We truly appreciate the reviewer’s valuable comments. We have carefully studied the reviewer’s comments and revised our manuscript accordingly.

“-The authors claim that displacement steps Δh are not power law distributed, while stress drops are. I can not see any convincing evidence for this. The "power laws" in Fig. S1 are as good (or as bad) as those in Fig. 2. I suggest to bin in logarithmic scale: this is conventionally done for power law distributions. Please also show bin-free distributions by reporting (in the supplement) the cumulative distribution by simply rank-ordering the recorded values of Δh (and $\Delta\sigma$).”

Thank you for the valuable comment and suggestion. We now understand that the displacement burst (displacement step) Δh is also power-law-distributed. In addition, owing to better statistical independence of Δh , as discussed later according to the valuable comment of reviewer #2, we decided to mainly use the Δh data for the subsequent pop-ins power law analyses within the statistical independent displacement range of $20.0-40.0 \text{ nm} \leq h \leq 90.0-145.0 \text{ nm}$ using bin-free cumulative distributions, as shown in revised Fig. 2, even though $\Delta\sigma$ can acceptably demonstrate the power law, as seen in Supplementary information S4. Moreover, according to the reviewer’s suggestion, we also show equal-width binning, logarithmic binning, and bin-free cumulative distributions side-by-side in Supplementary information S5. Because artifacts are not expected in the bin-free cumulative treatment, we decided to estimate the scaling exponent using the bin-free cumulative distribution. Note that for the first pop-in analysis, we also use Δh in addition to $\Delta\sigma$ to perform a “first-subsequent (all-data) mixed analysis,” as we discuss later.

“The authors make a complicated argument with several assumptions to measure stress drops. While all this might be reasonable, I would like to see a demonstration. This could be obtained by MD simulations where it is possible to have at the same time access to stresses and displacements. The authors should verify that their relation between stress drops and displacement is correct.”

Thank you for the valuable suggestions. We completely agree. Because the pop-in load $P^{\text{pop-in}}$ is constant before and after a pop-in, contact stress drop is “exactly” represented by Eq. (1). Thus, we only need to make sure whether the contact area just before and after a pop-in, $A(h)$ and $A(h+\Delta h)$. We confirmed the Hertz’s contact theory $A(h) = \pi R h$ before the first pop-in (elastic region) by MD simulation (please see Supplementary information S2). In MD simulation, there is no unique definition and computation method of contact area between two contacting objects owing to the “discrete” nature of MD simulation. Here, we simply, but reasonably, define the contact area $A(h)$ as $A(h) = N(h) \times \Omega / r_c$, where $r_c = 0.53$ nm is the potential cut-off distance between an indenter tip and atoms in the testing materials; Ω is the atomic volume in the bulk system ($=a^3 / n$, $n = 2$ (BCC) and $n = 4$ (FCC), where a is lattice constant), and $N(h)$ is the number of atoms in the volume contacting indenter tip, i.e., atoms within the cut-off distance from the indenter tip. Before the first pop-in, $A(h)$ actually follows $\pi R h$. In the revised manuscript, the experimental contact area at the end of the first pop-in, such as $A(h + \Delta h)$, and at the beginning and the end of the second and subsequent pop-ins, such as $A(h)$ and $A(h + \Delta h)$ respectively, are estimated by a quadratic area function; $A(h) = A(h_{\text{con}}(h)) = c_2(h_{\text{con}}(h))^2 + c_1 h_{\text{con}}(h) + c_{1/2}(h_{\text{con}}(h))^{1/2}$ ($c_2 = 24.5$, $c_1 = 2.61 \times 10^3$ nm, and $c_{1/2} = 1.57 \times 10^{-7}$ nm^{3/2}) considering the effect of the rounded indenter tip shape through Oliver-Pharr (OP) method [22] (W. C. Oliver and G. M. Pharr, J. Mater Res. 7, 1564 (1992)), where a contact depth $h_{\text{con}}(h)$ is given as $h_{\text{con}}(h) = h - P(h) / S(h)$, $P(h)$ is the measured load, $S(h) = (2/\sqrt{\pi})E_r\sqrt{A(h_{\text{con}}(h))}$ is an effective stiffness, and E_r is a reduced elastic modulus defined by using the Young’s modulus and Poisson’s ratio of both the target and the indenter materials (for the details, please see Methods and Supplementary information S3). The contact area $A(h)$ was obtained by numerically solving these equations. This is based on the idea that the material surface has a trigonal-pyramid-like shape, reflecting the shape of the indenter body because the indenter tip had already made a deep indent to the material. The area function has been calibrated preliminarily for our indenter tip by using a standard sample of fused silica through Oliver-Pharr method [22] (W. C. Oliver and G. M. Pharr, J. Mater Res. 7, 1564 (1992)) (see also Supplementary information S3).

“It is argued that MD simulations can not be compared with experiments because of the different loading mode. The argument is not very convincing. I do not understand the claim that displacement control was needed “to maintain stability”. One can certainly apply a force to the indenter and

slowly increase it and I can not see how this will affect the stability of the simulation. But even if there is a practical reason for which force driven simulations are unfeasible (and the authors should explain it with much more care), the authors should still do more to compare experiments and simulations: for instance they should measure the pop-in stress drop distribution from MD simulations. It is really a pity that almost non information is extracted from the MD simulations.”

We are sorry. The word “maintain stability” may lead to a misunderstanding. We wanted to say that even though we conducted the load-controlled MD nanoindentation simulation, it was difficult to compare the simulation results with the actual load control experiments owing to the dynamic parameter issue between MD and the experiment. When we perform the load-controlled MD, we need to consider the dynamics of the indenter tip motion. This is essential because when a pop-in occurs, the force acting on the indenter tip suddenly falls into a force-unbalanced condition, and indenter motion must be accelerated by the unbalanced force subject to the equation of motion. However, it is very difficult to estimate the apparent inertial mass of the indenter tip and the effective damping factor of the nanoindentation system, which are necessary to solve the equation of motion of the indenter tip; thus, we cannot uniquely determine the time-dependent indenter position without these parameters. Moreover, even if we obtain appropriate parameters, it is expected that MD simulation cannot demonstrate the actual dynamics of the indentation test owing to its extremely shorter time scale than that of the experiment. Meanwhile, even using the displacement-controlled MD, we cannot directly compare MD with experiments because we can directly obtain the force drop instead of the displacement burst, and we also have the time-scale issue through the displacement rate. Thus, both methods have issues. However, because in displacement-controlled MD we do not need to use the unknown parameters related to the indenter tip dynamics, we still use the displacement-controlled MD. We have revised the manuscript and added the above discussion.

In addition to the above discussion, according to the reviewer’s suggestion, we performed 60 independent displacement-controlled MD simulations for 1) BCC Fe (100) at 5 and 500 K to obtain the event distribution for the first pop-in, its temperature dependence, and the power-law exponent of the second and subsequent pop-ins at 5 K and 2) BCC Fe (111) at 5 K to obtain the power-law exponent of the second and subsequent pop-ins at 5 K; the results are given in Supplementary information S9. The MD simulation also demonstrates the Gaussian-like first pop-in distribution, its temperature dependence, and the power-law universal scaling behavior in subsequent pop-ins with exponents of $\beta = 5.0$ for (100) and 3.8 for (111) using a force drop distribution, which reasonably agree with the experimental exponents. Note that these MD distributions were plotted with respect to the force drop owing to the suggestion by Reviewer #1 in addition to those plotted with respect to the displacement burst. Moreover, we have added MD snapshots and movies for dislocation and

plastic strain nucleation and evolution and stress field and atomic displacement evolution beneath the indenter during the first and subsequent pop-ins to Figs. 3 and 4 (in the main text) and Supplementary information S15 and S16 with movies “Movies.pptx”).

*“-Are the Δh steps statistically independent on time? Clearly the first step is larger but are all the other steps statistically equivalent? To test this the authors could plot $\langle \Delta h | h \rangle$ vs. h (i.e. average step size at given displacement level) and/or $\langle \Delta h | k \rangle$ with k being the pop-in number (first, second, etc). Since $N=1000$ indentations are performed this should not be a problem. The same analysis should be performed for the stress drop. Doing this is extremely important to understand the universality class of the distribution. If the Δh (or $\Delta \sigma$) signal is not stationary then the high exponent of the distribution (i.e. $\beta=2.8$) can be explained as a result of the integration of a non-stationary signal. This is discussed in previous papers on crackling noise, see for instance Durin, G., & Zapperi, S. (2006). The role of stationarity in magnetic crackling noise. *Journal of Statistical Mechanics: Theory and Experiment*, 2006(01), P01002.”*

Thank you very much for your enormously fruitful comments. To confirm the statistical independence of the subsequent pop-ins data, we plotted $\langle \Delta h | h \rangle$ vs. h and $\langle \Delta \sigma | h \rangle$ vs. h (i.e., conditional probabilities; an average displacement burst and an average contact stress drop at given displacement h (indentation depth)), as shown in Supplementary information S10.

It is clear that Δh shows a nice statistical independence within a wide range of indentation depths; $20.0\text{--}40.0 \text{ nm} \leq h \leq 90.0\text{--}145.0 \text{ nm}$, while $\Delta \sigma$ shows a weak depth dependence, as seen in Fig. S10-2. Physically, the displacement burst Δh is directly related to the external work done by an indenter by assuming that a constant indentation load is maintained over the pop-in event, while the previously used stress drop $\Delta \sigma$ is based on the force-balanced states before and after the pop-in. Therefore, we decided to use the Δh data for the subsequent pop-in power law analyses within the statistical independent displacement range, even though $\Delta \sigma$ can acceptably demonstrate the power law, as seen in Figs. S4-1 in Supplementary information S4. The newly estimated power law exponents using Δh are $\beta = 5.6$ for the BCC Fe (100) surface and $\beta = 3.9$ for the BCC Fe (111) surface, as shown in the revised Fig. 2. Here, according to the comment by Reviewer #2 regarding data analysis, the exponents were calculated from the cumulative distributions, as shown in the revised Fig. 2. Moreover, in addition to BCC Fe, according to the Reviewer #1’s comment, we also performed nanoindentation testing for the FCC Cu (100) surface and obtained $\beta = 6.4$. Despite the revision, all power law exponents are still significantly higher than those of usual pillar compression testing ($\beta = 1.0\text{--}1.8$) and nanoindentation testing and simulation ($\beta = 1.5\text{--}1.6$) (Reference [20] (R.

Bolin et al., Crystal, 9, 1 (2019)) and Reference [19] (H. Song et al., J. Mech. Phys. Solids, 123, 332 (2019))).

Here, the question arises why our exponent $\beta = 6.4$ for FCC Cu is much higher than the exponents $\beta = 1.5-1.6$ for FCC metals, which have been reported in the nanoindentation experiments and simulation. After the following additional analysis “without” separating the first pop-in data [first-subsequent (all data) mixed analysis], we strongly believe that the difference is a result of whether the first pop-in is separated from the others. To demonstrate this, we replot Δh for both Fe and Cu (shown in Fig. S11-1 in Supplementary information S11) “without” separating the first pop-in data and using the same logarithmic binning analysis as in the nanoindentation paper (R. Bolin et al., Crystal, 9, 1 (2019)) for the same FCC Cu. We actually obtained a similar power law exponent $\beta \cong 1.6$, which has been reported in the nanoindentation papers (please see Fig. S11-1(c)). In addition, we would also like to emphasize that, especially for BCC Fe (Fig. S11-1(a)(b)), the “first-subsequent (all-data) mixed analysis” does not show the power law scaling at all because clear separation does exist between the first and subsequent pop-in magnitude distributions due to the higher ideal strain of BCC metals than FCC metals (Reference[28] S. Ogata et al., Phys. Rev. B, 70, 104104 (2004)), as seen in Fig. S11-1(a)(b). Moreover, the statistical independence analysis of Δh with respect to indentation depth h (please see Supplementary information S10) also suggests that the first pop-in must be different from the others. Hence, the additional analyses for FCC Cu and the “first-subsequent mixed analysis” provides us with the following insights: 1) the separation of the first several pop-ins from the others is necessary, and 2) the separation shows hidden high-exponent universal-scaling world in nanoindentation testing.

We have added the above discussions and figures to the manuscript and Supplementary information.

“-minor point: I do not understand the role of c in the x-axis of Fig. 2. The c constant is not discussed anywhere and looks completely useless (since of course c cancels out!).”

We apologize for the typographical error. We have revised Fig. 2.

Responses to Reviewer 3

“This manuscript describes a series of nanoindentation experiments performed on iron single crystals, and preliminary analysis of molecular dynamics simulations of the same process. The central claim is that the contact stress drop of the initial pop-in roughly follows a Gaussian distribution, whereas the contact stress drop of the subsequent pop-ins roughly follows a power law distribution. The exponent of the power law is empirically measured to be higher than those generally observed in micro-pillar plasticity. This is suggested as evidence that the phenomena belong to different universal scaling classes.”

Thank you very much for your careful reading of our manuscript and valuable comments.

Before addressing the reviewer’s comments, we would like to discuss revisions to the statistical analysis method for subsequent pop-ins. In the revised manuscript, we simply employed the displacement burst Δh , which is directly measured in indentation testing as the measure of the pop-in magnitude for the subsequent analysis of pop-ins instead of the stress drop $\Delta\sigma$, which has been previously used in the manuscript; we still use the stress drop for the first pop-in analysis. We made these changes because we received two valuable comments from Reviewer #2: 1) “ Δh seems to also follow the power law” and 2) “statistical independence with respect to the indentation depth h of the subsequent pop-in data should be checked.” To confirm comment 2), we plotted $\langle \Delta h | h \rangle$ vs. h and $\langle \Delta\sigma | h \rangle$ vs. h , i.e., conditional probabilities; an average displacement burst Δh and an average contact stress drop $\Delta\sigma$ at a given indentation depth h are shown in Figs. S10-1 and S10-2. It is clear that Δh exhibits a nice statistical independence within a wide range of indentation depth, 20.0–40.0 nm $\leq h \leq$ 90.0–145.0 nm, while $\Delta\sigma$ shows a weak depth dependence, as seen in Fig. S10-2. Physically, the displacement burst Δh is directly related to the external work done by an indenter under the assumption that the constant indentation load is maintained over the pop-in event, while the previously used stress drop $\Delta\sigma$ is based on the force balanced states before and after the pop-in. Therefore, we decided to use the Δh data for the subsequent pop-in power-law analyses within the statistical independent displacement range even though $\Delta\sigma$ can acceptably demonstrate the power law, as seen in Fig. S4-1 in Supplementary information S4. The newly estimated power law exponents using Δh are $\beta = 5.6$ for the BCC Fe (100) surface and $\beta = 3.9$ for the BCC Fe (111) surface, as shown in revised Fig. 2. Here, according to the comment from Reviewer #2 for data analysis, the exponents were calculated from cumulative distributions, as shown in revised Fig. 2. For reference, equal-width binning and logarithmic binning plots are also shown in Fig. S5-1. Moreover, in addition to BCC Fe, according to Reviewer #1’s valuable comment, we also performed

nanoindentation testing for the FCC Cu (100) surface and obtained $\beta = 6.4$. Nevertheless, all power law exponents are still significantly higher than those obtained by the usual pillar-compression testing, $\beta = 1.0\text{--}1.8$, and by nanoindentation testing and simulations, $\beta \cong 1.6$ ((Reference [20] (R. Bolin et al., *Crystal*, 9, 1 (2019)) and Reference [19] (H. Song et al., *J. Mech. Phys. Solids*, 123, 332 (2019))). Of note, for the first pop-in analysis, we also used Δh in addition to $\Delta\sigma$ to perform a “first-subsequent (all-data) mixed analysis,” as discussed later. Moreover, we developed a novel avalanche model (please see Supplementary information S13), which can successfully explain the origin of power-law exponents in the second and subsequent pop-ins, which is related to material properties and surface orientation, and can explain also the reason for the difference in the power law exponent between the micro-pillar compression and nanoindentation testing.

Before continuing with the techniques and analysis, I would like to discuss the significance of the claims. Whether or not the manuscript provides sufficient support for the claim that the physical process underlying the second and subsequent pop-ins belongs to a different universality class than that behind micro-pillar plasticity, it is unclear why this would be a result of general interest. I would certainly be interested in a precise explanation of the differences between these processes, particularly with regard to dislocation interaction and multiplication. That said, just reporting that they occur with different probability distributions of contact stresses does not substantially increase my understanding.

As for the question of universality class, the knowledge that two processes belong to the same universality class (in principle) allows the transfer of some understanding about one process to the other. That is not done here, with the claim of differing universality classes amounting to the tautological statement "different processes are different".

I do wish to acknowledge that the experiments are not trivial to perform, and that the authors have collected a substantial quantity of data about a question that could be of general interest. The analysis and conclusions feel incomplete though, and partly for that reason I do not believe that this manuscript would substantially change thinking in the field.”

Thank you very much for the insightful comments and suggestions to improve our manuscript.

We agree that the difference between the micro-pillar compression testing and the nanoindentation testing should be explained from the standpoint of dislocation activities to more clearly represent the underlying physics and reason for different universality classes.

The high exponents, and material and surface orientation dependencies may be attributed to the unique boundary condition and stress distribution of nanoindentation, which differ from

uniaxial-loading pillar compression testing. Here, we would like to recall that the displacement avalanche results from the motion of dislocation ensembles, and the magnitude of the avalanche is proportional to the total migration distance of dislocations.

The driving force of dislocation (more exactly it is “dislocation segment”; however simply “dislocation” is used hereafter for simplicity) in a material is the stress exerted on the dislocation, which originates from both the application of an external force to the material and the stress field produced by other dislocations. The fundamental difference between these testing methods is the force-applying geometry and the resulting stress field. In micro-pillar load-controlled compression testing (to a single crystal of pure metal), since the exerted stress distribution on a slip plane directly originated from applying an external force: “background stress distribution” should be uniform over the slip plane; once dislocation starts to move, the dislocation motion can only be suppressed and then terminated by a dislocation-dislocation interaction, such as the indirect elastic interaction via the stress field from the other dislocations and additional direct dislocation-dislocation reactive interactions. Thus, dislocation multiplication is the major termination mechanism of the displacement avalanche in micro-pillar compression testing (T. Crosby et al., *Acta Mater.*, 89, 123 (2015).), while dislocation escape from the free surface is another minor termination mechanism. Meanwhile, in extremely small nano-pillar testing, dislocation nucleation from the free surface, and then passing through the entire sample and escaping from the free surface can be the major mechanism because of the small probability of dislocation multiplication and thus the small possibility of dislocation-dislocation interaction (J. R. Greer et al., *Phys. Rev. B*, 73, 245410 (2006), J. Kim et al., *Int. J. Plast.*, 28, 46 (2012)). In this case, however, we should always observe a unique magnitude of displacement avalanche without any distribution, which is uniquely determined by the pillar diameter. This is not in our interest.

However, in nanoindentation testing, the background stress distribution produced by the indenter is not uniform (K. J. Van Vleet et al., *Phys. Rev. B*, 67, 104105 (2003), T. Zhu et al., *J. Mech. Phys. Solids*, 52, 691 (2004)). Although a local very high stress field (the level of which is comparable with the ideal shear strength (T. Zhu et al., *J. Mech. Phys. Solids*, 52, 691 (2004)). is created in the local region near the indenter tip, the stress level rapidly decreases with an increase in the distance from the indenter tip. Thus, the starvation of the driving force owing to the lack of background stress at remote fields far from the indenter tip is one of the reasons for the termination of the displacement burst, in addition to the direct dislocation-dislocation interaction owing to dislocation multiplication as we directly show using the MD results in the revised manuscript.

To more directly demonstrate the dislocation activities during the first and subsequent pop-ins

(during the displacement burst), we have carefully analyzed the MD simulation results from the standpoint of dislocation activities and stress field beneath the indenter. In addition to the visualization of dislocation pattern immediately before and after a pop-in and von Mises stress invariant distribution beneath the indenter, we also visualized the atomic-scale distributions of the change in the von Mises atomic-strain invariant (F. Shimizu et al., *Mater. Trans.*, 48, 2923 (2007)) during the first pop-in as shown in Fig. 3 for BCC Fe (100), S15-1 for BCC Fe (111), and S15-2 for FCC Cu (100) and those during subsequent pop-ins as shown in Fig. 4 for BCC Fe (100), S16-1 for BCC Fe (111), and S16-2 for FCC Cu (100)). The von Mises strain invariant visualization method allows us to directly observe the atoms contributing to the plastic deformation (displacement burst) produced within a pop-in. Moreover, together with dislocation pattern visualizations before and after the pop-in, the history of dislocation motion during the pop-in can be also determined. The von Mises atomic-strain invariant was computed on the basis of the atomic displacement of each atom from the beginning to the end of the pop-in. A cutoff strain of 0.1 was set to avoid the detection of change in elastic strain, which typically must be smaller than 0.1. We also visualized atomic displacement along indenter axis to see which regions are contributing to the indenter displacement.

At the first pop-in, vast dislocations were nucleated and spread in a fanlike pattern from the indenter tip (Fig. 3(a)(b)), which have been observed in a nanoindentation experiment using transmission electron microscopy (TEM) (L. Zhang et al., *Phys. Rev. Lett.*, 112, 145504 (2014)), eventually a local high-density dislocation field was formed in the near field of the indenter tip. At the same time, the stress distribution beneath the indenter immediately contracted and decreased (Fig. 3(c)(d)), and then, the indentation load dropped (Fig. 3(e)) with generating a plastic strain (Fig. 3(f)) and atomic displacement (Fig. 3(g)) mostly at just beneath the indenter tip. After that, the decreasing trend of the indentation load shifted to an increasing trend. Then, after recovering the indentation load, the second and, then, the subsequent pop-ins may also occur. During the subsequent pop-in, the dislocation field expanded out, and simultaneously new dislocations were formed at the local high stress field near the indenter tip (Fig. 4(a)(b)), with reducing the total stress distribution beneath the indenter (Fig. 4(c)(d)) by consuming the elastic strain energy stored before the pop-in starts and work done by indenter during the pop-in, even though there was no gain of the background stress (rather a decrease). At the same time, some of dislocations existing at the remote fields (pointed by purple arrow in Fig. 4 (a)(b)) traveled further owing to an additional “pushing” force originated from the near field dislocation motions [= a reactive force to the backstress force acting on the near field dislocations from the remote field dislocations] (dislocation motion cascade). These dislocation activities generated a certain amount of plastic strain (Fig. 4(f)) and atomic displacements (Fig. 4(g)) with contributing to the indenter displacement. However, this pushing force does not increase forever because the following dislocations also lose the driving force for these motions when they

come into the remote fields with a lower background stress. Moreover, the background stress distribution itself decreases under the constant pop-in load because of a decrease of the contact stress (stress drop) between the indenter and the target material owing to the increase of contact area with the progress of the pop-in. Thus, the dislocations existing in the remote fields will be stopped. Actually, as seen in Fig. 4 and Figs. S16-1 and S16-2, some of the remote field dislocations stopped without escaping from the surface at the bottom of the model. The near-field dislocations exhibited a vigorous activity with generating plastic strain to accommodate the indenter tip motion, which were nucleated in the subsequent pop-in at the very local high stress field beneath the indenter. These dislocations directly interact with each other owing to the high local dislocation density, and most of them seem to become an immobile soon after generating a certain amount of plastic strain and indenter displacement with the reduction of the remote field stress, while some of them (pointed by orange arrow in Fig. 4(a)(b)) escaped out to the remote fields with the generation of indenter displacement.

At this stage, the near field dislocations can no longer contribute to the generation of further plastic strain and thus the indenter displacement either through its motion or by pushing other dislocations, while the escaped dislocations contributed to the indenter displacement, and, then, stopped after a certain travel from the near field up to balancing the backstress from the remote field dislocations, pushing force from the near field dislocations, and the background stress. All of the abovementioned unique termination mechanisms of the dislocation motion in nanoindentation testing originate from the nonuniform stress distribution with a rapid decay with respect to the distance from the indenter tip, in addition to the dislocation-dislocation direct interaction in the near field. Hence, the pop-in owing to the dislocation avalanche is fundamentally restricted by the unique stress distribution of nanoindentation testing via restricting the dislocation motion, which is not formed in the micro-pillar compression testing. The additional restriction is the reason for the high power-law exponents, i.e., the probability of large-scale events becomes significantly small.

We have added the above discussion to the manuscript.

Again, on the basis of the abovementioned discussion and MD observations regarding dislocation activities in the unique nanoindentation stress field, we developed a novel avalanche model (please see Supplementary information S13). The model can successfully explain the origin of the power-law exponents in the second and subsequent pop-ins, which is related to materials intrinsic properties, temperature, and the surface property. In addition, the model can explain the reason for the difference in the power law exponent between micro-pillar compression and nanoindentation testing.

Regarding the power law experiment on FCC nanoindentation, it is important to understand why our exponent $\beta = 6.4$ for FCC Cu is much higher than exponents $\beta = 1.5\text{--}1.6$ for FCC metals, which have been reported in nanoindentation experiments and simulation (Reference [20] (R. Bolin et al., *Crystal*, 9, 1 (2019)) and Reference [19] (H. Song et al., *J. Mech. Phys. Solids*, 123, 332 (2019)) cited in our manuscript). After the following additional analysis “without” separating the first pop-in data (first–subsequent (all data) mixed analysis), we strongly believe that the difference is a result of whether the first pop-in is separated from others. To demonstrate this, we re-plot Δh for both Fe and Cu, as shown in Fig. S11-1 (in Supplementary information S11), “without” separating the first pop-in data and using the same logarithmic binning analysis as in the nanoindentation papers for the same FCC Cu. Actually, we obtained a similar power law exponent $\beta \cong 1.6$ as the one reported in previous nanoindentation papers (please see Fig. S11-1(c)). In addition, we also emphasize that especially for BCC Fe (Fig. S11-1 (a)(b)), the “first-subsequent mixed analysis” does not show the power law scaling because a clear separation does exist between the first and subsequent pop-in magnitude distributions, as seen in Fig. S11-1 (a)(b). Moreover, the statistical independence analysis of Δh with respect to indentation depth h (please see Supplementary information S10) suggests that the first pop-in must be different from the others. Hence, the additional analyses for FCC Cu and the “first-subsequent mixed analysis” allow us to make the following conclusions: 1) the separation of the first several pop-ins from other pop-ins is necessary, and 2) the separation unveils a hidden high-exponent universal-scaling-world in nanoindentation testing.

We have added the above discussions and figures to the manuscript and Supplementary information.

We believe that the analysis and discussion, which are based on MD simulations and the new dislocation avalanche model, as well as the first–subsequent mixed analysis can change the thinking of the universality class of plasticity in nanoindentation; thus, the exponent is not always approximately $\beta = 1.6$ as generally believed; the exponent can be much higher under an extremely local stress-concentrated field with a high stress gradient, which can be realized in nanoindentation testing but not in micro-pillar testing.

“I have several concerns with the techniques and analysis. These are enumerated below:

1. The definition of a second pop-in is not precise. The authors set an arbitrary velocity threshold above which the event is identified as a pop-in, but this is unsatisfactory for several reasons. First, have the authors verified that the events they sample are not simply electrical and mechanical noise? The noise in electrical instruments is often pink noise, which could explain the observed power law probability distribution they observe in Figure 2. One natural resolution would be to consider all

events after the first pop-in, and to see whether there is a natural decomposition of the resulting probability distribution into a noise contribution for the smallest events and everything else. Second, the main characteristic of the first pop-in is that there is irreversible plastic deformation during the event, but not before. The unloading curves in Figure 1 indicate that there is irreversible plastic deformation all along the loading curve after the first pop-in. Then are the points identified as subsequent pop-ins at all distinguishable from the rest of the loading curve on physical grounds?"

Thank you for pointing out this important point. Concerning the first point, to investigate displacement difference owing to electrical and mechanical noise (which mainly consists of many types of noise including pink noise with $1/f$ distribution and white noise with flat and Gaussian distributions), the noise floor in the captured data acquired at our facilities should be analyzed using the probability function. To separate the noise floor from the essential displacement by plastic deformation, contiguous displacement data should be recorded in a constant applied load with the same capture rate. Because mechanical noise may depend on the applied load, different peak load conditions should be set. We conducted additional nanoindentation experiments on the (100) surface of BCC Fe by holding three types of peak indentation loads (10, 100, and 1000 μN) for 100 s and recorded the displacement, as shown in Fig. S1-1 (Supplementary information S1). The loading rate of the indenter was 50 $\mu\text{N/s}$, which is the same as that in the nanoindentation experiments described in the manuscript, and the sampling rate was 200 points per second. Figure S1-2 shows the magnified curve in the nanoindentation experiment when holding the peak load at 10 μN . There is an irregular variation even under the constant load condition, which is due to the nanoindentation device by thermal drift. For each case of peak loads, the displacement differences between adjacent points during the holding period of 100 s are plotted and shown in Fig. S1-3. The number of captured data is approximately 20,000 for each condition. Almost all the displacement differences are less than ± 0.4 nm. Because the threshold value in the manuscript was 0.5 nm, we assume that the displacement differences owing to only the noise are not included in the results in the manuscript. However, it is possible that the detected displacement burst includes the noise part. Unfortunately, even if we remove the frequency of the noise by conducting frequency analysis on the basis of discrete Fourier transformation, the pop-ins and noise cannot be separated because plastic deformation events are likely to occur within shorter time than the resolution of the analysis. According to the probability distribution of the noise size (shown in Fig. S1-4), the obtained noise is Gaussian noise for any peak load condition, which is completely different from pink noise. Therefore, the noise floor in our system does not include pink noise, and the indentation pop-in data in the manuscript mainly correspond to the mechanical reaction to the applied stress.

Regarding the second point, the red arrows that we put as subsequent pop-ins on the load–

displacement curve in Fig. 1 were just an example. To avoid such confusion, we revised Fig. 1 and show all of the detected second and subsequent pop-ins. As the reviewer has pointed out, we have many small subsequent events (other than the utilized events) that satisfy the velocity threshold. Unfortunately, we cannot physically distinguish them; we can only distinguish them with a threshold that is based on S/N and/or measurement limits. Thus, probability distributions have a small-event truncation owing to the non-physical threshold. However, such threshold should be used even for the micro-pillar plasticity analysis.

“2. Perhaps the difficulty with identifying discrete subsequent pop-ins is that there is effectively continuous thermal activation of dislocation-related events. This could be suppressed by increasing the indentation rate and decreasing the temperature, thereby strengthening the claim that the subsequent pop-ins really are significant events. This could also have the effect of allowing the subsequent pop-ins to be more directly related to the athermal activation energies, allowing a more fundamental characterization of the phenomenon. Where is the temperature reported?”

Thank you for your valuable comment. First, we apologize for not clearly stating the temperature in the manuscript. It was room temperature, i.e., 300 K. We have stated the temperature and loading rate 50 $\mu\text{N/s}$ in the revised manuscript. We completely agree with the reviewer’s comment that the use of more athermal conditions is more ideal, such as the “free-from-thermal-noise” avalanche behavior in subsequent pop-ins. Unfortunately, owing to the technical limitations of our indentation-testing machine, we cannot lower the temperature. However, an increase in temperature is possible. Thus, we conducted additional nanoindentation experiments on the (100) surface of BCC Fe at 373 K in addition to the room temperature at the same 50 $\mu\text{N/s}$ loading rate. The obtained distributions of subsequent pop-in magnitude are shown in Fig. S12-1 in Supplementary information S12. Although the power law exponent with a decrease in the temperature, as we can expect on the basis of our dislocation avalanche model (please see again Supplementary information S13), the power law distributions can be confirmed even at the higher temperature, moreover it can be suggested that lower temperature leads to a higher power-law exponent. Furthermore, we performed 60 independent displacement-controlled MD simulations for BCC Fe (100) and BCC Fe (111) at a very low temperature 5 K and a very high loading rate 5 m/s to obtain the power-law exponent of the second and subsequent pop-ins at around athermal limit; the results are given in Supplementary information S9. Our MD simulations also exhibit a power law scaling and similar high power-law exponents to those of experiments, such as $\beta = 5.0$ and 3.8.

Hence, we believe that our power-law analysis using the data at the loading rate of 50 $\mu\text{N/s}$

demonstrates the essence of the universal scaling of the subsequent pop-ins; therefore, we expect that at a higher loading rate, which is closer to athermal conditions, the exponent may become rather larger but not considerably change.

We have added the above discussion to the manuscript.

“3. Given the importance of the contact area to the calculation of the stress drop magnitudes, it is surprising and concerning that the authors did not verify the assumption that the contact area is proportional to the square of the indentation depth. Such verification would be relatively straightforward, involving a sequence of interrupted tests and subsequent investigation of the surface profile.”

We apologize for not clearly stating this information in the original manuscript, although we have already verified the quadratic area function of the geometry of the indenter based on the standard Oliver-Pharr (OP) method (W. C. Oliver and G. M. Pharr, *J. Mater Res.* 7, 1564 (1992)). The details are described in Supplementary information 3 as the following description.

The contact area $A(h_{\text{con}}(h))$ can be basically calculated by the contact depth $h_{\text{con}}(h)$ through the indenter geometry. There are problems for getting accurate values in both parameters. One is the estimation of the $h_{\text{con}}(h)$ by separation from elastic displacement in measured depth h , the other is calibration of an imperfection in the indenter geometry. The problems are generally solved through the “area function” in O-P method, which has been referred in several works for a variety of materials (W. C. Oliver and G. M. Pharr, *J. Mater Res.* 7, 1564 (1992)).

For the former problem, the contact depth $h_{\text{con}}(h)$ is hard to be calculated because the measured depth h at peak load includes both plastic and elastic displacements. Therefore, the estimation of the $h_{\text{con}}(h)$ by separation from the elastic displacement strongly impinges on an accuracy of the contact area evaluation. The h_{con} can be estimated by an analysis of unloading curve with purely elastic recovery from a peak load as shown schematically in Fig. S3-1, which is typically obtained for fused silica as the standard sample. When we have a hysteresis loop of load-depth curve including the segments of loading and subsequent unloading, some parameters are obtained as follows. P_{max} and h_{max} are the measured load and penetration depth at the peak load, respectively. If a contact area between the indenter and sample at P_{max} is kept constant during unloading, a flat-end punch model is applicable to get a load-depth relation and then the unloading segment should be rectilinear. The linear line can be drawn with the slope $S = S(h_{\text{max}})$ that is the unloading stiffness analytically given as a differential coefficient at the peak load of a power function fitted with the unloading curve. By

extrapolating the linear line to the horizontal axis, the cross point is obtained as the contact depth h_{con} , which corresponds to the penetration depth within the contact region between the indenter and sample at peak load and is analytically calculated as

$$h_{\text{con}} = h_{\text{max}} - \frac{P_{\text{max}}}{S(h_{\text{max}})}. \quad (1)$$

Note that the really measured unloading curve generally shows a curved line as shown in Fig. S3-1, which means a gradual decreasing in the contact area due to an elastic recovery inside the contact region during unloading, leading to the final penetration depth h_{fin} after getting back to zero in load. Therefore, the h_{fin} is smaller than h_{con} and gives us a wrong value in the contact area at the peak load, and h_{con} should be used instead.

For the later problem, we can calibrate the indenter geometry by using the standard sample of fused silica. In the case of an ideal shape of Berkovich indenter in the three-sided pyramid with apex angle of 115° , the contact area $A(h_{\text{con}}(h))$ is given as $A(h_{\text{con}}(h)) = 24.5 h_{\text{con}}(h)^2$. However, a real tip shape is generally not perfect but truncated by an ablation, and hence the shape should be calibrated. In the OP model, the $S(h)$ is given as a function of $A(h_{\text{con}}(h))$,

$$S(h) = \frac{2}{\sqrt{\pi}} E_r \sqrt{A(h_{\text{con}}(h))}, \quad (2)$$

where E_r is a reduced elastic modulus and can be written for isotropic material as

$$E_r^{-1} = (1-\nu_i^2)/E_i + (1-\nu_s^2)/E_s, \quad (3)$$

where E and ν are Young's modulus and Poisson's ratio, and subscripts i and s refer to indenter and sample, respectively. For fused silica, E_r is 70 GPa (W. C. Oliver and G. M. Pharr, J. Mater Res. 7, 1564 (1992)). Therefore, the $A(h_{\text{con}}(h_{\text{max}}))$ is calculated by the measurement of $S(h_{\text{max}})$ through Eq. (2) for the P_{max} . When we vary the P_{max} for $P_{\text{max}}^1 \dots P_{\text{max}}^i \dots P_{\text{max}}^n$ as shown in Fig. S3-1, we can obtain the corresponding $A(h_{\text{con}}(h_{\text{max}}^1)) \dots A(h_{\text{con}}(h_{\text{max}}^i)) \dots A(h_{\text{con}}(h_{\text{max}}^n))$. Also, the corresponding $h_{\text{con}}(h_{\text{max}}^1) \dots h_{\text{con}}(h_{\text{max}}^i) \dots h_{\text{con}}(h_{\text{max}}^n)$ can be obtained through Eq. S3-1. By plotting the $A(h_{\text{con}}(h))$ vs. $h_{\text{con}}(h)$, we can obtain the area function by fitting to the standard quadratic function given in

$$A(h_{\text{con}}) = c_2 h_{\text{con}}^2 + c_1 h_{\text{con}} + c_{1/2} h_{\text{con}}^{1/2}, \quad (4)$$

where $c_2 = 24.5$, $c_1 = 2.61 \times 10^3$ nm, and $c_{1/2} = 1.57 \times 10^{-7}$ nm^{3/2}. The second and third terms in the quadratic area function are the correction term for the imperfect tip. Once we get the area function of the indenter geometry, it can be used for every material universally, and E_r can be evaluate through a measurement of $S(h)$.

For making sure that the quadratic area function works for the Fe single crystal, we conducted additional nanoindentation experiments with setting four different peak loads (250, 500, 750 and 1000 μ N) on the (100) surface of the BCC Fe. We measured $A'(h_{\text{fin}}(h))$ as an contact area by a direct measurement on the atomic force microscope (AFM) image instead of $S(h)$ measurement for cross checking, and then $A'(h_{\text{fin}}(h))$ is compared with $A(h_{\text{con}}(h))$ that is independently given by $h_{\text{con}}(h)$ through the quadratic area function in Eq. (4). Note that the area function $A'(h_{\text{fin}}(h))$ is different from $A(h_{\text{con}}(h))$ because the geometry of the indent print is different due to the elastic recovery within the contact region, but both functions follow the quadratic function basically. The AFM images of the surface imprint in each nanoindentation are shown in the following Fig. S3-2. The image is presented in a gradient mode with 256×256 pixels², in which the contrast corresponds to a local cant on the surface. The triangle imprints are clearly shown and the three vertexes or sides of the triangle are determined. The triangles size increases as the peak load increases. However, the top view image may be not suitable for measuring the contact area because the residual imprint may have a different geometry from the indenter due to an elastic recovery inside the indent imprint. Therefore, the cross-section profiles of the imprints were measured as Figure S3-3. On the profile, the origin in the vertical axis is the original height of the sample surface. We can determine the edge of the triangle as the red and green cross positions. The distance between them can be measured as a representative horizontal size. Also, the maximum depth of the imprint is given in the profile, which corresponds to the $h_{\text{fin}}(h)$ in Fig. S3-1. By using the values of horizontal and vertical sizes, the contact area $A'(h_{\text{fin}}(h)) = ch_{\text{fin}}(h)^2$ can be calculated based on an assumption of regular triangular pyramid for the four load conditions. The plot of the independently calculated $A'(h_{\text{fin}}(h))$ and $A(h_{\text{con}}(h))$ are shown in Fig. S3-4. The plots fit well with a linear line, meaning the area function in the quadratic function works in Fe. It should be noted that the $A'(h_{\text{fin}}(h))$ is lower about 5 % than $A(h_{\text{con}}(h))$ with the same P_{max} condition because the $A(h_{\text{con}}(h))$ includes an elastic displacement under the maximum load while $A'(h_{\text{fin}}(h))$ is in the condition after unloading without any elastic deformation.

When we calculate $\Delta\sigma$ at the end of the first pop-in, at the beginning and at the end of the second and subsequent pop-in, we used $A(h_{\text{con}}(h))$ in Eq. (1) for a strict evaluation. To get the $A(h_{\text{con}}(h))$ though Eq. (1), the $S(h)$ is necessary by unloading analysis. Since the $S(h)$ cannot be measured in the case on a continuous loading segment without unloading, the $A(h_{\text{con}}(h))$ is numerically solved on the

combination of Eqs. (1) and (2) by using only P as P_{\max} and h as h_{\max} on the loading segment. Note that the E_r for Fe and Cu samples in Eq. (2) is evaluated in each sample as given in method section. Although the above analysis clearly shows that the ch^2 relationship is reasonable, just in case, in the revised manuscript we used a more accurate quadratic area function instead of ch^2 .

For further making sure the contact area in the Fe sample, the area function is calculated using independently measured $S(h)$ and $h_{\text{con}}(h)$ in the four load conditions as $A(h) = (S(h)\pi/2E_r)^2$ vs. $h_{\text{con}}(h)$ in the following figure. The E_r of 200 GPa was used as rough value from the measured average. The solid line shows the area function in Eq. (4). The plots fit well with the solid line, confirming the area function for the Fe sample.

Figure: The plot $A(h) = (S(h)\pi/2E_r)^2$ vs. $h_{\text{con}}(h)$ to make sure the area function in Eq. (4) for the Fe sample.

“4. The authors do not appear to address the possible formation of an oxide layer on the surface of the Fe. Such an oxide layer could form very quickly indeed, and be related to the difference between the first pop-in (film rupture) and subsequent pop-ins (dislocation multiplication). What assurance can the authors provide that such an oxide layer is not present?”

Thank you for pointing out this important point. To investigate the oxide layer formation on the surface of Fe, we measured the thickness of the oxide layer using a spectroscopic ellipsometer (MARY-102FM, Five Lab, Yokohama, Japan). The instrument was operated by the rotating retarder

method and used a 0.8-mW HeNe laser with a single wavelength of 632.8 nm and a spot diameter of 0.8 mm. The incident beam angle was set as 69.97° , and the reflected beam angle was measured with a resolution of $\pm 0.01^\circ$. The beam scanned the sample surface in an area of $1 \times 1 \text{ mm}^2$ with a 0.25-mm step, which resulting in 5×5 positions. To measure the oxide layer thickness, we assumed that the oxide layer was either Fe_2O_3 or Fe_3O_4 with refractive indices of 2.918 or 2.42, respectively. The 2D map of the measured thickness of the Fe_2O_3 layer is shown in Fig. S17-1 in Supplementary information S17. The discrete data are connected with a spline curve with a 10-pixel resolution, which results in $40 \times 40 \text{ pixels}^2$. The map shows a gradual distribution in thickness with a maximum value of 6.15 nm and minimum value of 4.98 nm; the average thickness was determined to be about 5.49 nm. We conclude that this thickness is within the negligible range compared to the radius of the indenter R ($\sim 500 \text{ nm}$). Because the oxide layer may be much harder than iron, the oxide layer acts as a “pseudo” indenter tip for iron when a load is applied by a “real” diamond indenter to the surface oxide. On the basis of the Hertz contact theory, the maximum shear stress under the indenter is inversely proportional to $R^{2/3}$, and 1% deviation in R corresponds to the 0.67% difference in the shear stress, which is absolutely within experimental error.

“5. Have the authors considered fitting the probability distribution of initial pop-ins using the well-established model of Schuh and Lund (DOI: 10.1557/JMR.2004.0276)? What is the advantage of the model derived in the supplementary material over that one?”

Thank you for your comment. We regret not referring to the Schuh and Lund model. Actually, the concept of our model is basically the same as the of Schuh and Lund model; however, in our study, the probability is more directly represented as a function of the pop-in load $P^{\text{pop-in}}$ instead of the local shear stress at the dislocation nucleation point.

We have clearly stated this in the revised manuscript.

“6. The simulations do not provide any atomistic insight into the processes occurring during the so-called subsequent pop-ins, and have been performed many times before in the literature. It is unclear what value the simulations add to the manuscript.”

We agree with the reviewer that the contribution of simulation in the original manuscript is small. First, the atomic insights into the process occurring in the subsequent pop-in have been already discussed at the beginning of this reply. We hope the reviewer agrees that the simulation provides

important atomistic insights into the process occurring during the subsequent pop-ins.

In addition, the 60 independent displacement-controlled MD simulations for both BCC Fe (100) at 5 and 500 K and (111) at 5 K shown in Supplementary information S9 demonstrates the Gaussian-like first pop-in distribution and its temperature dependence as well as, again, the power law universal scaling behavior in subsequent pop-ins with the exponents of $\beta = 5.0$ for (100) and 3.8 for (111) using the force drop distribution, which reasonably agree with the experimental exponents. Note that these MD distributions are plotted with respect to the force drops owing to the suggestion of Reviewer #1, in addition to those plotted with respect to the displacement burst. Moreover, we have added MD snapshots and movies for dislocation and plastic strain nucleation and evolution and stress field and atomic displacement evolution beneath the indenter during the first and subsequent pop-ins to Figs. 3 and 4 (in the main text) and Supplementary information S15 and S16 with movies “Movies.pptx.”

“7. Why does the power-law exponent reported here differ so substantially from that reported in Ref. 19? That the power-law exponent in Ref. 19 is entirely consistent with the exponent for micro-pillar plasticity suggests that this manuscript is the outlier.”

Thank you for your valuable comment. As we mentioned above, we estimated the power law exponents with and without the first pop-in data, which are $\beta \cong 1.6$ and $\beta = 6.4$, respectively. The former is consistent with exponents reported in nanoindentation plasticity experiments and simulation, $\beta \cong 1.6$ (Reference [20] (R. Bolin et al., Crystal 2019) and Reference [19] cited in our manuscript); thus, we believe that high exponent can be uncovered only using the subsequent pop-in statistics by separating the first pop-in data from those of other pop-ins.

“8. What is the algorithm that the authors use for the fitting of the models? I don't see any mention of this, or the resulting uncertainties in the fitting parameters.”

Thank you for your comment. Regarding the fitting of the first pop-in distribution, we set the initial fitting parameter by trial and error owing to strong nonlinearity; then, we used the typical Newton–Raphson method. We added relevant information to the Supplementary information S6. Regarding the fitting of the subsequent pop-in distribution, we used the least squares fitting method, as described in the manuscript.

Reviewers' Comments:

Reviewer #1:

Remarks to the Author:

I believe that the authors meticulously addressed all comments, inquiries and suggestions that all the referees posed. I believe the work is an excellent contribution to the field and should be published in Nature Communications.

Reviewer #2:

Remarks to the Author:

The authors have work very hard to answer the constructive remarks of the referees considerably improving their manuscript. I am satisfied by the corrections and suggest to accept the paper in the present form.

Reviewer #3:

Remarks to the Author:

Manuscript ID: NCOMMS-19-41235

Title of Article: Unique universal scaling in nanoindentation pop-ins

Since this is the second round of reviews, I will dispense with the customary summary of the manuscript contents.

The authors have made a substantial effort to address my earlier concerns, and in particular have greatly expanded the analysis of the MD results and the discussion of what properties of the dislocation arrangements could lead to the observed power-law scaling. I now believe that, while this does not report on any scientific breakthrough of general interest, the research is well-done and should attract the attention of those concerned with the mechanics of materials and avalanche processes.

I still have several concerns which are enumerated below:

1. The language can be difficult to follow in places. I trust that the editorial staff would be able to address these difficulties though.
2. Some of the added material does not feel very focused. In particular, the comparison with the micropillar experiments in p11 and the discussion of the MD simulations on p14,15 could be made more concise. This would not only shorten the manuscript, but make the author's points clearer.
3. Some of the material that was added to Results should be moved to the Methods section. For example, the discussion of the area function could be moved. Similarly, the discussion of displacement vs load control in the MD simulations should be moved to the Methods section.
4. There appears to be a missing factor of g in the exponent in Eq. 5 of S13, and the authors should consider replacing Eq. 5 with an integral of the log probability. Then the passage to the logarithm in Eq. 6 could be made exact.

Responses to Reviewer 3

“The authors have made a substantial effort to address my earlier concerns, and in particular have greatly expanded the analysis of the MD results and the discussion of what properties of the dislocation arrangements could lead to the observed power-law scaling. I now believe that, while this does not report on any scientific breakthrough of general interest, the research is well-done and should attract the attention of those concerned with the mechanics of materials and avalanche processes.”

Thank you for your comment. Following your comments, we have considerably improved the quality of the manuscript. We really appreciate your careful peer reviewing.

“I still have several concerns which are enumerated below:

1. The language can be difficult to follow in places. I trust that the editorial staff would be able to address these difficulties though.”

We are sorry for making linguistic mistakes. We have requested a proofreading of the manuscript.

“2. Some of the added material does not feel very focused. In particular, the comparison with the micropillar experiments in p11 and the discussion of the MD simulations on p14,15 could be make more concise. This would not only shorten the manuscript, but make the author's points clearer.”

Thank you for your valuable comment. According to the reviewer’s comment, we have revised the comparison with the micropillar experiments on page 11 and the discussion of the MD simulations on pages 14 and 15.

“3. Some of the material that was added to Results should be moved to the Methods section. For example, the discussion of the area function could be moved. Similarly, the discussion of displacement vs load control in the MD simulations should be moved to

the Methods section.”

Thank you for your valuable comment. According to the reviewer’s comment, we have moved the discussion of the area function and displacement vs. load control in MD simulations to Page 18, line 5 and Page 19, line 25 in the Methods section, respectively.

“4. There appears to be a missing factor of g in the exponent in Eq. 5 of S13, and the authors should consider replacing Eq. 5 with an integral of the log probability. Then the passage to the logarithm in Eq. 6 could be made exact.”

We appreciate your careful reading. Before addressing your comment, we revised the equation numbers as “S1, S2, ...” sequentially throughout the Supplementary Information to avoid confusion between equations in the main text and in the Supplementary Information.

Thank you for your comment; we did find an error regarding factor g . Indeed, we should not have the factor g in front of kT in Eq. (S27) [Eq. (4) in the previous version]. In particular, there was an extra g in Eq. (S27); therefore, Eq. (S28) [Eq. (5) in the previous version] is correct. Equation (S27) was corrected in the revised manuscript. Additionally, following your advice, the summation in the exponential function was replaced by an integral. We chose to do it in Eq. (S29) [Eq. (6) in the previous version] and leave the summation in Eq. (S28) so that readers could follow the calculation more easily.